# Spherical Sliced-Wasserstein

## Abstract

Many variants of the Wasserstein distance have been introduced to reduce its
original computational burden. In particular the Sliced-Wasserstein distance (SW),
which leverages one-dimensional projections for which a closed-form solution
of the Wasserstein distance is available, has received a lot of interest. Yet, it is
restricted to data living in Euclidean spaces, while the Wasserstein distance has
been studied and used recently on manifolds. We focus more specifically on the
sphere, for which we define a novel SW discrepancy, which we call spherical Sliced-
Wasserstein, making a first step towards defining SW discrepancies on manifolds.
Our construction is notably based on closed-form solutions of the Wasserstein
distance on the circle, together with a new spherical Radon transform. Along
with efficient algorithms and the corresponding implementations, we illustrate its
properties in several machine learning use cases where spherical representations
of data are at stake: density estimation on the sphere, variational inference or
hyperspherical auto-encoders.

## 1   Introduction

Optimal transport (OT) [101] has received a lot of attention in machine learning in the past few years.
As it allows to compare distributions with metrics, it has been used for different tasks such as domain
adaptation [24] or generative models [8], to name a few. The most classical distance used in OT is
the Wasserstein distance. However, calculating it can be computationally expensive. Hence, several
variants were proposed to alleviate the computational burden, such as the entropic regularization
[26, 97], minibatch OT [35] or the sliced-Wasserstein distance (SW) for distributions supported on
Euclidean spaces [90].

Although embedded in larger dimensional Euclidean spaces, data generally lie in practice on manifolds
[36]. A simple manifold, but with lots of practical applications, is the hypersphere $S^{d-1}$. Several
types of data are by essence spherical: a good example is found in directional data [71, 87] for
which dedicated machine learning solutions are being developed [98], but other applications concern
for instance geophysical data [32], meteorology [11], cosmology [86] or extreme value theory
for the estimation of spectral measures [44]. Remarkably, in a more abstract setting, considering
hyperspherical latent representations of data is becoming more and more common (*e.g.* [28, 70, 110]).
For example, in the context of variational autoencoders [58], using priors on the sphere has been
demonstrated to be beneficial [28]. Also, in the context of self-supervised learning (SSL), where
one wants to learn discriminative representations in an unsupervised way, the hypersphere is usually
considered for the latent representation [20, 21, 43, 104, 108]. It is thus of primary importance to
develop machine learning tools that accommodate well with this specific geometry.

Submitted to 36th Conference on Neural Information Processing Systems (NeurIPS 2022). Do not distribute.

The OT theory on manifolds is well developed [38, 73, 101] and several works started to use it in practice, with a focus mainly on the approximation of OT maps. For example, Cohen et al. [23], Rezende and Racanière [91] approximate the OT map to define normalizing flows on Riemannian manifolds, Cui et al. [25], Hamfeldt and Turnquist [45, 46] derive algorithms to approximate the OT map on the sphere, Alvarez-Melis et al. [5], Hoyos-Idrobo [51] learn the transport map on hyperbolic spaces. However, the computational bottleneck to compute the Wasserstein distance on such spaces remains, and, as underlined in the conclusion of [74], defining SW distances on manifolds would be of much interest.

**Contributions.** Therefore, by leveraging properties of the Wasserstein distance on the circle [89], we define the first, to the best of our knowledge, SW discrepancy on a non trivial manifold, namely the sphere $S^{d-1}$, and hence we make a first step towards defining SW distances on Riemannian manifolds. We make connections with a new spherical Radon transform and analyze some of its properties. We discuss the underlying algorithmic procedure, and notably provide an efficient implementation when computing the discrepancy against a uniform distribution. Then, we show that we can use this discrepancy on different tasks such as density estimation, variational inference or generative modeling.

## 2 Background

The aim of this paper is to define a Sliced-Wasserstein discrepancy on the hypersphere $S^{d-1} = \{x \in \mathbb{R}^d, \|x\|_2 = 1\}$. Therefore, in this section, we introduce the Wasserstein distance on manifolds and the classical SW distance on $\mathbb{R}^d$.

### 2.1 Wasserstein distance

Since we are interested in defining a SW discrepancy on the sphere, we start by introducing the Wasserstein distance on a Riemannian manifold $M$ endowed with the Riemannian distance $d$. We refer to [38, 101] for more details.

Let $p \geq 1$ and $\mu, \nu \in \mathcal{P}_p(M) = \{\mu \in \mathcal{P}(M), \int_M d^p(x, x_0) \, \mathrm{d}\mu(x) < \infty \text{ for some } x_0 \in M\}$. Then, the $p$-Wasserstein distance between $\mu$ and $\nu$ is defined as

$$W_p^p(\mu, \nu) = \inf_{\gamma \in \Pi(\mu,\nu)} \int_{M \times M} d^p(x, y) \, \mathrm{d}\gamma(x, y), \tag{1}$$

where $\Pi(\mu, \nu) = \{\gamma \in \mathcal{P}(M \times M), \ \forall A \subset M, \ \gamma(M \times A) = \nu(A) \text{ and } \gamma(A \times M) = \mu(A)\}$ denotes the set of couplings.

For discrete probability measures, the Wasserstein distance can be computed using linear programs [88]. However, these algorithms have a $O(n^3 \log n)$ complexity *w.r.t.* the number of samples $n$ which is computationally intensive. Therefore, a whole literature consists of defining alternative discrepancies which are cheaper to compute. On Euclidean spaces, one of them is the Sliced-Wasserstein distance.

### 2.2 Sliced-Wasserstein distance

On $M = \mathbb{R}^d$ with $d(x, y) = \|x - y\|_p^p$, a more attractive distance is the Sliced-Wasserstein (SW) distance. This distance relies on the appealing fact that for one dimensional measures $\mu, \nu \in \mathcal{P}(\mathbb{R})$, we have the following closed-form [88, Remark 2.30]

$$W_p^p(\mu, \nu) = \int_0^1 \left| F_\mu^{-1}(u) - F_\nu^{-1}(u) \right|^p \, \mathrm{d}u, \tag{2}$$

where $F_\mu^{-1}$ (resp. $F_\nu^{-1}$) is the quantile function of $\mu$ (resp. $\nu$). From this property, Bonnotte [16], Rabin et al. [90] defined the SW distance as

$$\forall \mu, \nu \in \mathcal{P}_p(\mathbb{R}^d), \ SW_p^p(\mu, \nu) = \int_{S^{d-1}} W_p^p(P_\#^\theta \mu, P_\#^\theta \nu) \, \mathrm{d}\lambda(\theta), \tag{3}$$

where $P^\theta(x) = \langle x, \theta \rangle$, $\lambda$ is the uniform distribution on $S^{d-1}$ and for any Borel set $A \in \mathcal{B}(\mathbb{R}^d)$, $P^\theta_\# \mu(A) = \mu((P^\theta)^{-1}(A))$.

This distance can be approximated efficiently by using a Monte-Carlo approximation [75], and amounts to a complexity of $O(Ln \log n)$ where $L$ denotes the number of projections used for the Monte-Carlo approximation and $n$ the number of samples.

SW can also be written through the Radon transform [15]. Let $f \in L^1(\mathbb{R}^d)$, then the Radon transform $R : L^1(\mathbb{R}^d) \to L^1(\mathbb{R} \times S^{d-1})$ is defined as [48]

$$\forall \theta \in S^{d-1}, \ \forall t \in \mathbb{R}, \ Rf(t, \theta) = \int_{\mathbb{R}^d} f(x) \mathbb{1}_{\{\langle x, \theta \rangle = t\}} \mathrm{d}x. \tag{4}$$

Its dual $R^* : C_0(\mathbb{R} \times S^{d-1}) \to C_0(\mathbb{R}^d)$ (also known as back-projection operator), where $C_0$ denotes the set of continuous functions that vanish at infinity, satisfies for all $f, g$, $\langle Rf, g \rangle_{\mathbb{R} \times S^{d-1}} = \langle f, R^* g \rangle_{\mathbb{R}^d}$ and can be defined as [13, 15]

$$\forall g \in C_0(\mathbb{R} \times S^{d-1}), \forall x \in \mathbb{R}^d, \ R^* g(x) = \int_{S^{d-1}} g(\langle x, \theta \rangle, \theta) \, \mathrm{d}\theta. \tag{5}$$

Therefore, by duality, we can define the Radon transform of a measure $\mu \in \mathcal{M}(\mathbb{R}^d)$ as the measure $R\mu \in \mathcal{M}(\mathbb{R} \times S^{d-1})$ such that for all $g \in C_0(\mathbb{R} \times S^{d-1})$, $\langle R\mu, g \rangle_{\mathbb{R} \times S^{d-1}} = \langle \mu, R^* g \rangle_{\mathbb{R}^d}$. Since $R\mu$ is a measure on the product space $\mathbb{R} \times S^{d-1}$, we can disintegrate it $w.r.t.$ $\lambda$, the uniform measure on $S^{d-1}$ [6], as $R\mu = \lambda \otimes K$ with $K$ a probability kernel on $S^{d-1} \times \mathcal{B}(\mathbb{R})$, $i.e.$ for all $\theta \in S^{d-1}$, $K(\theta, \cdot)$ is a probability on $\mathbb{R}$, for any Borel set $A \in \mathcal{B}(\mathbb{R})$, $K(\cdot, A)$ is measurable, and

$$\forall \phi \in C(\mathbb{R} \times S^{d-1}), \ \int_{\mathbb{R} \times S^{d-1}} \phi(t, \theta) \mathrm{d}(R\mu)(t, \theta) = \int_{S^{d-1}} \int_{\mathbb{R}} \phi(t, \theta) K(\theta, \mathrm{d}t) \mathrm{d}\lambda(\theta). \tag{6}$$

By Proposition 6 in [15], we have that for $\lambda$-almost every $\theta \in S^{d-1}$, $(R\mu)^\theta = P^\theta_\# \mu$ where we denote $K(\theta, \cdot) = (R\mu)^\theta$. Therefore, we have

$$\forall \mu, \nu \in \mathcal{P}_p(\mathbb{R}^d), \ SW_p^p(\mu, \nu) = \int_{S^{d-1}} W_p^p\big((R\mu)^\theta, (R\mu)^\theta\big) \, \mathrm{d}\lambda(\theta). \tag{7}$$

Variants of SW have been defined in recent works, either by integrating $w.r.t.$ different distributions [31, 80, 81], by projecting on $\mathbb{R}$ using different projections [78, 79] or Radon transforms [22, 60], or by projecting on subspaces of higher dimensions [52, 66, 67, 85].

# 3 A Sliced-Wasserstein discrepancy on the sphere

Our goal here is to define a sliced-Wasserstein distance on the sphere $S^{d-1}$. To that aim, we proceed analogously to the classical Euclidean space. We first rely on the nice properties of the Wasserstein distance on the circle [89] and then propose to project distributions lying on the sphere to great circles. Hence, circles play the role of the real line for the hypersphere. In this section, we first describe the OT problem on the circle, then we define a sliced-Wasserstein discrepancy on the sphere and discuss some of its properties. Notably, we derive a new spherical Radon transform which is linked to our newly defined spherical SW. We refer to Appendix A for the proofs.

## 3.1 Optimal transport on the circle

On the circle $S^1 = \mathbb{R}/\mathbb{Z}$ equipped with the geodesic distance $d_{S^1}$, an appealing formulation of the Wasserstein distance is available [30]. First, let us parametrize $S^1$ by $[0, 1[$, then the geodesic distance can be written as [89], for all $x, y \in [0, 1[$, $d_{S^1}(x, y) = \min(|x - y|, 1 - |x - y|)$. Then, for the cost function $c(x, y) = h(d_{S^1}(x, y))$ with $h : \mathbb{R} \to \mathbb{R}^+$ an increasing convex function, the Wasserstein distance between $\mu \in \mathcal{P}(S^1)$ and $\nu \in \mathcal{P}(S^1)$ can be written as

$$W_c(\mu, \nu) = \inf_{\alpha \in \mathbb{R}} \int_0^1 h\big(|F_\mu^{-1}(t) - (F_\nu - \alpha)^{-1}(t)|\big) \, \mathrm{d}t, \tag{8}$$

where $F_\mu : [0,1[ \to [0,1]$ denotes the cumulative distribution function (cdf) of $\mu$, $F_\mu^{-1}$ its quantile function and $\alpha$ is a shift parameter. The optimization problem over the shifted cdf $F_\nu - \alpha$ can be seen as looking for the best "cut" (or origin) of the circle into the real line because of the 1-periodicity. Indeed, the proof of this result for discrete distributions in [89] consists in cutting the circle at the optimal point and wrapping it around the real line, for which the optimal transport map is the increasing rearrangement $F_\nu^{-1} \circ F_\mu$ which can be obtained for discrete distributions by sorting the points [88].

Rabin et al. [89] showed that the minimization problem is convex and coercive in the shift parameter and Delon et al. [30] derived a binary search algorithm to find it. For the particular case of $h = \mathrm{Id}$, it can further be shown [19, 106] that

$$W_1(\mu, \nu) = \inf_{\alpha \in \mathbb{R}} \int_0^1 |F_\mu(t) - F_\nu(t) - \alpha| \, \mathrm{d}t. \tag{9}$$

In this case, we know exactly the minimum which is attained at the level median [53]. For $f : [0,1[ \to \mathbb{R}$,

$$\mathrm{LevMed}(f) = \min \left\{ \operatorname*{argmin}_{\alpha \in \mathbb{R}} \int_0^1 |f(t) - \alpha| \mathrm{d}t \right\} = \inf \left\{ t \in \mathbb{R}, \ \beta(\{x \in [0,1[, \ f(x) \le t\}) \ge \frac{1}{2} \right\}, \tag{10}$$

where $\beta$ is the Lebesgue measure. Therefore, we also have

$$W_1(\mu, \nu) = \int_0^1 |F_\mu(t) - F_\nu(t) - \mathrm{LevMed}(F_\mu - F_\nu)| \, \mathrm{d}t. \tag{11}$$

Since we know the minimum, we do not need the binary search and we can approximate the integral very efficiently as we only need to sort the samples to compute the level median and the cdfs.

Another interesting setting in practice is to compute $W_2$, *i.e.* with $h(x) = x^2$, *w.r.t.* a uniform distribution $\nu$ on the circle. We derive here the optimal shift $\hat{\alpha}$ for the Wasserstein distance between $\mu$ an arbitrary distribution on $S^1$ and $\nu$. We also provide a closed-form when $\mu$ is a discrete distribution.

**Proposition 1.** *Let $\mu \in \mathcal{P}_2(S^1)$ and $\nu = \mathrm{Unif}(S^1)$. Then,*

$$W_2^2(\mu, \nu) = \int_0^1 |F_\mu^{-1}(t) - t - \hat{\alpha}|^2 \, \mathrm{d}t \quad with \quad \hat{\alpha} = \int x \, \mathrm{d}\mu(x) - \frac{1}{2}. \tag{12}$$

*In particular, if $x_1 < \cdots < x_n$ and $\mu_n = \frac{1}{n} \sum_{i=1}^n \delta_{x_i}$, then*

$$W_2^2(\mu_n, \nu) = \frac{1}{n} \sum_{i=1}^n x_i^2 - \left( \frac{1}{n} \sum_{i=1}^n x_i \right)^2 + \frac{1}{n^2} \sum_{i=1}^n (n+1-2i)x_i + \frac{1}{12}. \tag{13}$$

This proposition offers an intuitive interpretation: the optimal cut point between an empirical and a uniform distributions is the antipodal point of the circular mean of the discrete samples. Moreover, a very efficient algorithm can be derived from this property, as it solely requires a sorting operation to compute the order statistics of the samples.

## 3.2 Definition of SW on the sphere

On the hypersphere, the counterpart of straight lines are the great circles, which correspond to the geodesics. Moreover, we can compute the Wasserstein distance on the circle fairly efficiently. Hence, to define a sliced-Wasserstein discrepancy on this manifold, we propose, analogously to the classical SW distance, to project measures on great circles. The most natural way to project points from $S^{d-1}$ to a great circle $C$ is to use the geodesic projection [40, 55] defined as

$$\forall x \in S^{d-1}, \ P^C(x) = \operatorname*{argmin}_{y \in C} d_{S^{d-1}}(x, y), \tag{14}$$

where $d_{S^{d-1}}(x, y) = \arccos(\langle x, y \rangle)$ is the geodesic distance. See Figure 1 for an illustration of the geodesic projection on a great circle. Note that the projection is unique for almost every $x$ (see

[9, Proposition 4.2] and Appendix B.1) and hence the pushforward $P_\#^C \mu$ of absolutely continuous measures *w.r.t.* the Lebesgue measure $\mu \in \mathcal{P}_{p,ac}(S^{d-1})$ is well defined.

Great circles can be obtained by intersecting $S^{d-1}$ with a 2-dimensional plane [56]. Therefore, to average over all great circles, we propose to integrate over the Grassmann manifold $\mathcal{G}_{d,2} = \{E \subset \mathbb{R}^d, \dim(E) = 2\}$ [2, 10] and then to project the distribution onto the intersection with the hypersphere. Since the Grassmannian is not very practical, we consider the identification using the set of rank 2 projectors:

$$\mathcal{G}_{d,2} = \{P \in \mathbb{R}^{d\times d}, \ P^T = P, \ P^2 = P, \ \mathrm{Tr}(P) = 2\} = \{UU^T, \ U \in \mathbb{V}_{d,2}\}, \tag{15}$$

where $\mathbb{V}_{d,2} = \{U \in \mathbb{R}^{d\times 2}, \ U^T U = I_2\}$ is the Stiefel manifold [10].

Finally, we can define the Spherical Sliced-Wasserstein distance (SSW) for $p \geq 1$ between locally absolutely continuous measures *w.r.t.* the Lebesgue measure [9] $\mu, \nu \in \mathcal{P}_{p,ac}(S^{d-1})$ as

$$SSW_p^p(\mu, \nu) = \int_{\mathbb{V}_{d,2}} W_p^p(P_\#^U \mu, P_\#^U \nu) \, d\sigma(U), \tag{16}$$

where $\sigma$ is the uniform distribution over the Stiefel manifold $\mathbb{V}_{d,2}$, $P^U$ is the geodesic projection on the great circle generated by $U$ and then projected on $S^1$, *i.e.*

$$\forall U \in \mathbb{V}_{d,2}, \forall x \in S^{d-1}, \ P^U(x) = U^T \underset{y \in \mathrm{span}(UU^T) \cap S^{d-1}}{\mathrm{argmin}} d_{S^{d-1}}(x, y) = \underset{z \in S^1}{\mathrm{argmin}} \ d_{S^{d-1}}(x, Uz), \tag{17}$$

and the Wasserstein distance is defined with the geodesic distance $d_{S^1}$.

Moreover, we can derive a closed form expression which will be very useful in practice:

**Lemma 1.** *Let $U \in \mathbb{V}_{d,2}$ then for a.e. $x \in S^{d-1}$,*

$$P^U(x) = \frac{U^T x}{\|U^T x\|_2}. \tag{18}$$

Hence, we notice from this expression of the projection that we recover almost the same formula as Lin et al. [66] but with an additional $\ell^2$ normalization which projects the data on the circle. As in [66], we could project on a higher dimensional subsphere by integrating over $\mathbb{V}_{d,k}$ with $k \geq 2$. However, we would lose the computational efficiency provided by the properties of the Wasserstein distance on the circle.

### 3.3 A Spherical Radon Transform

As for the classical SW distance, we can derive a second formulation using a Radon transform. Let $f \in L^1(S^{d-1})$, we define a spherical Radon transform $\tilde{R} : L^1(S^{d-1}) \to L^1(S^1 \times \mathbb{V}_{d,2})$ as

Figure 1: Illustration of the geodesic projections on a great circle (in black). In red, random points sampled on the sphere. In green the projections and in blue the trajectories.

$$\forall z \in S^1, \ \forall U \in \mathbb{V}_{d,2}, \ \tilde{R}f(z, U) = \int_{S^{d-1}} f(x) \mathbb{1}_{\{z = P^U(x)\}} dx. \tag{19}$$

This is basically the same formulation as the classical Radon transform [48, 77] where we replaced the real line coordinate $t$ by the coordinate on the circle $z$ and the projection is the geodesic one which is well suited to the sphere. This transform is actually new since we integrate over different sets compared to existing works on spherical Radon transforms.

Then, analogously to the classical Radon transform, we can define the back-projection operator $\tilde{R}^* : C_0(S^1 \times \mathbb{V}_{d,2}) \to C_b(S^{d-1})$, $C_b(S^{d-1})$ being the space of continuous bounded functions, for $g \in C_0(S^1 \times \mathbb{V}_{d,2})$ as for a.e. $x \in S^{d-1}$,

$$\tilde{R}^* g(x) = \int_{\mathbb{V}_{d,2}} g(P^U(x), U) \, d\sigma(U). \tag{20}$$

**Proposition 2.** $\tilde{R}^*$ *is the dual operator of* $\tilde{R}$, i.e. *for all* $f \in L^1(S^{d-1})$, $g \in C_0(S^1 \times \mathbb{V}_{d,2})$,

$$\langle \tilde{R}f, g \rangle_{S^1 \times \mathbb{V}_{d,2}} = \langle f, \tilde{R}^*g \rangle_{S^{d-1}}. \tag{21}$$

Now that we have a dual operator, we can also define the Radon transform of an absolutely continuous measure $\mu \in \mathcal{M}_{ac}(S^{d-1})$ by duality [13, 15] as the measure $\tilde{R}\mu$ satisfying

$$\forall g \in C_0(S^1 \times \mathbb{V}_{d,2}), \int_{S^1 \times \mathbb{V}_{d,2}} g(z, U) \, \mathrm{d}(\tilde{R}\mu)(z, U) = \int_{S^{d-1}} \tilde{R}^*g(x) \, \mathrm{d}\mu(x). \tag{22}$$

Since $\tilde{R}\mu$ is a measure on the product space $S^1 \times \mathbb{V}_{d,2}$, $\tilde{R}\mu$ can be disintegrated [6, Theorem 5.3.1] *w.r.t.* $\sigma$ as $\tilde{R}\mu = \sigma \otimes K$ where $K$ is a probability kernel on $\mathbb{V}_{d,2} \times \mathcal{S}^1$ with $\mathcal{S}^1$ the Borel $\sigma$-field of $S^1$. We will denote for $\sigma$-almost every $U \in \mathbb{V}_{d,2}$, $(\tilde{R}\mu)^U = K(U, \cdot)$ the conditional probability.

**Proposition 3.** *Let* $\mu \in \mathcal{M}_{ac}(S^{d-1})$, *then for* $\sigma$-*almost every* $U \in \mathbb{V}_{d,2}$, $(\tilde{R}\mu)^U = P_{\#}^U \mu$.

Finally, we can write SSW (16) using this Radon transform:

$$\forall \mu, \nu \in \mathcal{P}_{p,ac}(S^{d-1}), \ SSW_p^p(\mu, \nu) = \int_{\mathbb{V}_{d,2}} W_p^p\big((\tilde{R}\mu)^U, (\tilde{R}\nu)^U\big) \, \mathrm{d}\sigma(U). \tag{23}$$

Note that a natural way to define SW distances can be through already known Radon transforms using the formulation (23). It is for example what was done in [60] using generalized Radon transforms [34, 50] to define generalized SW distances, or in [22] with the spatial Radon transform. However, for known spherical Radon transforms [1, 7] such as the Minkowski-Funk transform [27] or more generally the geodesic Radon transform [95], there is no natural way that we know of to integrate over some product space and allowing to define a SW distance using disintegration.

As observed by Kolouri et al. [60] for the generalized SW distances (GSW), studying the injectivity of the related Radon transforms allows to study the set on which SW is actually a distance. While the classical Radon transform integrates over hyperplanes of $\mathbb{R}^d$, the generalized Radon transform over hypersurfaces [60] and the Minkowski-Funk transform over "big circles", *i.e.* the intersection between a hyperplane and $S^{d-1}$ [96], the set of integration here is a half of a big circle. Hence, $\tilde{R}$ is related to the hemispherical transform [94] on $S^{d-2}$. We refer to Appendix A.6 for more details on the links with the hemispherical transform. Using these connections, we can derive the kernel of $\tilde{R}$ as the set of even measures which are null over all hyperplanes intersected with $S^{d-1}$.

**Proposition 4.** $\ker(\tilde{R}) = \{\mu \in \mathcal{M}_{\text{even}}(S^{d-1}), \ \forall H \in \mathcal{G}_{d,d-1}, \ \mu(H \cap S^{d-1}) = 0\}$ *where* $\mu \in \mathcal{M}_{\text{even}}$ *if for all* $f \in C(S^{d-1})$, $\langle \mu, f \rangle = \langle \mu, f_+ \rangle$ *with* $f_+(x) = (f(x) + f(-x))/2$ *for all* $x$.

We leave for future works checking whether this set is null or not. Hence, we conclude here that SSW is a pseudo-distance, but a distance on the sets of injectivity of $\tilde{R}$ [4].

**Proposition 5.** *Let* $p \geq 1$, $SSW_p$ *is a pseudo-distance on* $\mathcal{P}_{p,ac}(S^{d-1})$.

## 4 Implementation

In practice, we approximate the distributions with empirical approximations and, as for the classical SW distance, we rely on the Monte-Carlo approximation of the integral on $\mathbb{V}_{d,2}$. We first need to sample from the uniform distribution $\sigma \in \mathcal{P}(\mathbb{V}_{d,2})$. This can be done by first constructing $Z \in \mathbb{R}^{d \times 2}$ by drawing each of its component from the standard normal distribution $\mathcal{N}(0, 1)$ and then applying the QR decomposition [67]. Once we have $(U_\ell)_{\ell=1}^L \sim \sigma$, we project the samples on the circle $S^1$ by applying Lemma 1 and we compute the coordinates on the circle using the $\mathrm{atan2}$ function. Finally, we can compute the Wasserstein distance on the circle by either applying the binary search algorithm of [30] or the level median formulation (11) for $SSW_1$. In the particular case in which we want to compute $SSW_2$ between a measure $\mu$ and the uniform measure on the sphere $\nu = \mathrm{Unif}(S^{d-1})$, we can use the appealing fact that the projection of $\nu$ on the circle is uniform, *i.e.* $P_{\#}^U \nu = \mathrm{Unif}(S^1)$ (particular case of Theorem 3.1 in [55], see Appendix B.3). Hence, we can use the Proposition 1 to compute $W_2$, which allows a very efficient implementation either by the closed-form (13) or approximation by rectangle method of (12). This will be of particular interest for applications in Section 5 such as autoencoders. We sum up the procedure in Algorithm 1.

---
**Algorithm 1** SSW
---

**Input:** $(x_i)_{i=1}^n \sim \mu$, $(y_j)_{j=1}^m \sim \nu$, $L$ the number of projections, $p$ the order
**for** $\ell = 1$ **to** $L$ **do**
    Draw a random matrix $Z \in \mathbb{R}^{d \times 2}$ with for all $i, j$, $Z_{i,j} \sim \mathcal{N}(0, 1)$
    $U = \mathrm{QR}(Z) \sim \sigma$
    Project on $S^1$ the points: $\forall i, j$, $\hat{x}_i^\ell = \frac{U^T x_i}{\|U^T x_i\|_2}$, $\hat{y}_j^\ell = \frac{U^T y_j}{\|U^T y_j\|_2}$
    Compute the coordinates on the circle $S^1$: $\forall i, j$, $\tilde{x}_i^\ell = (\pi + \mathrm{atan2}(-x_{i,2}, -x_{i,1}))/(2\pi)$, $\tilde{y}_j^\ell = (\pi + \mathrm{atan2}(-y_{j,2}, -y_{j,1}))/(2\pi)$
    Compute $W_p^p(\frac{1}{n}\sum_{i=1}^n \delta_{\tilde{x}_i^\ell}, \frac{1}{m}\sum_{j=1}^m \delta_{\tilde{y}_j^\ell})$ by binary search or (11) for $p = 1$
**end for**
Return $SSW_p^p(\mu, \nu) \approx \frac{1}{L}\sum_{\ell=1}^L W_p^p(\frac{1}{n}\sum_{i=1}^n \delta_{\tilde{x}_i^\ell}, \frac{1}{m}\sum_{j=1}^m \delta_{\tilde{y}_j^\ell})$

---

**Complexity.** Let us note $n$ (resp. $m$) the number of samples of $\mu$ (resp. $\nu$), and $L$ the number of projections. First, we need to compute the QR factorization of $L$ matrices of size $d \times 2$. This can be done in $O(Ld)$ by using *e.g.* Householder reflections [42, Chapter 5.2] or the Scharwz-Rutishauser algorithm [41]. Projecting the points on $S^1$ by Lemma 1 is in $O((n + m)dL)$ since we need to compute $L(n + m)$ products between $U_\ell^T \in \mathbb{R}^{2 \times d}$ and $x \in \mathbb{R}^d$. For the binary search or particular case formula (11) and (13), we need first to sort the points. But the binary search also adds a cost of $O((n + m)\log(\frac{1}{\epsilon}))$ to approximate the solution with precision $\epsilon$ [30] and the computation of the level median requires to sort $(n + m)$ points. Hence, for the general $SSW_p$, the complexity is $O(L(n + m)(d + \log(\frac{1}{\epsilon})) + Ln\log n + Lm\log m)$ versus $O(L(n + m)(d + \log(n + m)))$ for $SSW_1$ with the level median and $O(Ln(d + \log n))$ for $SSW_2$ against a uniform with the particular advantage that we do not need uniform samples in this case.

**Runtime Comparison.** We perform here some runtime comparisons. Using Pytorch [83], we implemented the binary search algorithm of [30] and used it with $\epsilon = 10^{-6}$. We also implemented $SSW_1$ using the level median formula (11) and $SSW_2$ against a uniform measure (12). All experiments are conducted on GPU.

On Figure 2, we compare the runtime between two distributions on $S^2$ between SSW, the Wasserstein distance and the entropic approximation using the Sinkhorn algorithm [26] with the geodesic distance as cost function. The distributions were approximated using $n \in \{10^2, 10^3, 10^4, 5 \cdot 10^4, 10^5\}$ samples of each distribution and we report the mean over 20 computations. We use the Python Optimal Transport (POT) library [39] to compute the Wasserstein distance and the entropic approximation. For large enough batches, we observe that SSW is much faster than its Wasserstein counterpart, and it also scales better in term of memory because of the need to store the $n \times n$ cost matrix. For small batches, the computation of SSW actually takes longer because of the computation of the QR factorizations and of the projections. For bigger batches, it is bounded

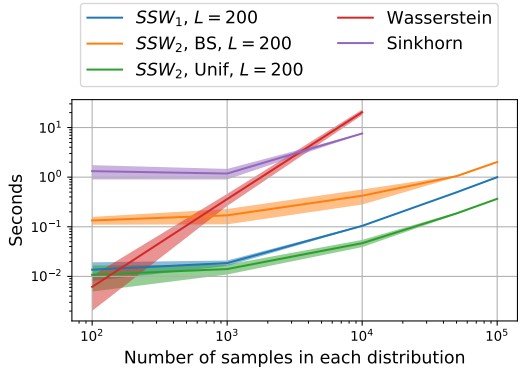

Figure 2: Runtime comparison in log-log scale between W, Sinkhorn with the geodesic distance, $SSW_2$ with the binary search (BS) and uniform distribution (12) and $SSW_1$ with formula (11) between two distributions on $S^2$. The time includes the calculation of the distance matrices.

by the sorting operation and we recover the quasi-linear slope. Furthermore, as expected, the fastest algorithms are $SSW_1$ with the level median and $SSW_2$ against a uniform as they have a quasilinear complexity. We report in Appendix C.2 other runtimes experiments *w.r.t.* to *e.g.* the number of projections or the dimension.

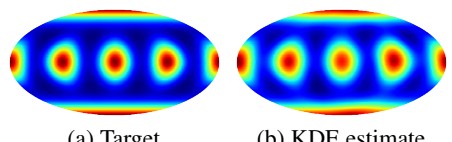
(a) Target      (b) KDE estimate

Figure 3: Minimization of SSW with respect to a mixture of vMF.

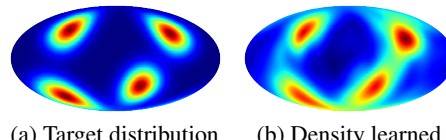
(a) Target distribution      (b) Density learned

Figure 4: Amortized SSWVI with a normalizing flow *w.r.t.* a mixture of vMF.

## 5 Applications

In this section, we first illustrate the ability to approximate different distributions by minimizing SSW *w.r.t.* some target distributions on $S^2$. We first use distributions from which we can draw samples. Then, we use target distributions from which we know the density only up to a constant. Finally, we apply SSW for generative modeling tasks using the framework of Sliced-Wasserstein autoencoder and we show that we obtain competitive results with other Wasserstein autoencoder based methods using a prior on the hypersphere. We also add in Appendix C.6 some experiments where we use SSW in order to enforce uniformity in a contrastive self-supervised learning context.

### 5.1 SSW as a loss

We verify on the two first experiments that we can learn some target distribution $\nu \in \mathcal{P}(S^{d-1})$ by minimizing SSW, *i.e.* we consider the minimization problem $\mathrm{argmin}_\mu \ SSW_p^p(\mu, \nu)$.

**Gradient flow.** First, we suppose that we have access to the target distribution $\nu$ through samples, *i.e.* through $\hat{\nu}_m = \frac{1}{m} \sum_{j=1}^m \delta_{y_j}$ where $(y_j)_{j=1}^m$ are i.i.d samples of $\nu$. We choose as target distribution a mixture of 6 well separated von Mises-Fisher distributions [72]. This is a fairly challenging distribution since there are 6 modes which are not connected. We show on Figure 3 the Mollweide projection of the density approximated by a kernel density estimator for a distribution with 500 particles. To optimize directly over particles, we can either perform a Riemannian gradient descent on the sphere [3] or a projected gradient descent. We report in Appendix C.3 additional details and experiments.

**Sliced-Wasserstein variational inference on the sphere.** Another setting of interest is when we have access to some target distribution only up to a constant. For example in Bayesian inference, we want to sample from a posterior distribution $p(\cdot|x)$ for which the normalizing constant is costly to compute, *i.e.* we can only evaluate some function $\pi$ such that $p(\cdot|x) \propto \pi$. Popular methods to solve these types of problems are MCMCs [93] or variational inference [12, 54].

Variational inference aims at approximating the target by a distribution $q$ in some family of distributions $\mathcal{Q}$. The classical way of doing it is to minimize the Kullback-Leibler (KL) divergence. However, the KL divergence suffers from some drawbacks such as under estimating the target distribution and not being a distance. Recently, Yi and Liu [111] proposed to use the SW distance instead. The method is called Sliced-Wasserstein Variation Inference (SWVI) and relies on running at each iteration few MCMC steps and then performing gradient descent to learn the variational distribution. We refer to Appendix C.4 and Algorithm 2 for further details on the method.

In the following, we replace SW by SSW in SWVI, which we denote SSWVI, and we perform amortized variational inference on the sphere by using exponential map normalizing flows (see [92] and Appendix B.4) to learn the distribution and the Geodesic Langevin algorithm [105] as MCMC method. We use the same target as Rezende et al. [92] and we report on Figure 4 the Mollweide projection of the learned density. Since we learn to sample from a noise distribution, here the uniform distribution on the sphere, we do not have directly access to the density and we report a kernel density estimate with a Gaussian kernel. On Figure 5, we plot the evolution of the effective samples size (ESS) [33, 69] through the iterations. This indicates how well the flow matches the target. We observe that using SSW gives slightly better results, or at least comparable, than SWVI with SW.

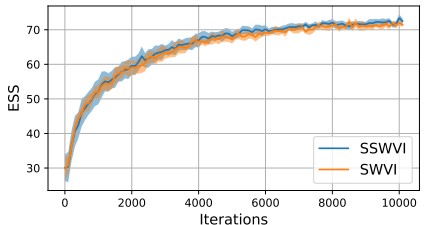

Figure 5: Comparison of the ESS between SWVI et SSWVI with the mixture target (mean and 95% confidence interval over 10 runs).

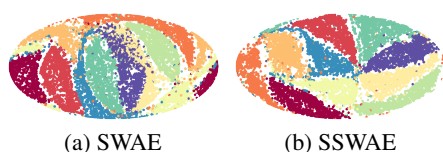

(a) SWAE  (b) SSWAE

Figure 6: Latent space of SWAE and SSWAE for a uniform prior on $S^2$.

## 5.2 SSW autoencoders

In this section, we use SSW to learn the latent space of autoencoders (AE). We rely on the SWAE framework introduced by Kolouri et al. [59]. Let $f$ be some encoder and $g$ be some decoder, denote $p_Z$ a prior distribution, then the loss minimized in SWAE is

Table 1: FID (Lower is better).

| Method / Dataset | MNIST | Fashion |
|---|---|---|
| SSWAE | **14.91 ± 0.32** | **43.94 ± 0.81** |
| SWAE | 15.18 ± 0.32 | 44.78 ± 1.07 |
| WAE-MMD IMQ | 18.12 ± 0.62 | 68.51 ± 2.76 |
| WAE-MMD RBF | 20.09 ± 1.42 | 70.58 ± 1.75 |
| SAE | 19.39 ± 0.56 | 56.75 ± 1.7 |
| Circular GSWAE | 15.01 ± 0.26 | 44.65 ± 1.2 |

$$\mathcal{L}(f,g) = \int c\big(x, g(f(x))\big) \mathrm{d}\mu(x) + \lambda SW_2^2(f_\#\mu, p_Z),$$
(24)

where $\mu$ is the distribution of the data for which we have access to samples. One advantage of this framework over more classical VAEs [58] is that no parametrization trick is needed here and therefore the choice of the prior is more free.

In several concomitant works, it was shown that using a prior on the hypersphere can improve the results [28, 110]. Hence, we propose in the same fashion as [59, 60, 84] to replace SW by SSW, which we denote SSWAE, and to enforce a prior on the sphere. In the following, we use the MNIST [64] and FashionMNIST [109] datasets, and we put an $\ell^2$ normalization at the output of the encoder. As a prior, we use the uniform distribution on $S^{10}$ and we compare in Table 1 the Fréchet Inception Distance (FID) [49], for 10000 samples and averaged over 5 trainings, obtained with the Wasserstein Autoencoder (WAE) [99], the classical SWAE [59], the Sinkhorn Autoencoder (SAE) [84] and circular GSWAE [60]. We observe that we obtain fairly competitive results. We add on Figure 6 the latent space obtained with a uniform prior on $S^2$. We observe a better separation between classes for SSWAE. We refer to appendix C.5 for more details and additional experiments.

## 6 Conclusion and discussion

In this work, we derive a new sliced-Wasserstein discrepancy on the hypersphere, that comes with practical advantages when computing optimal transport distances on hyperspherical data. We notably showed that it is competitive or even sometimes better than other metrics defined directly on $\mathbb{R}^d$ on a variety of machine learning tasks, including density estimation, variational inference or generative models. Our work is, up to our knowledge, the first to adapt the sliced Wasserstein framework to non-trivial manifolds. The three main ingredients are: *i)* a closed-form for Wasserstein on the circle, *ii)* a closed-form solution to the projection onto great circles, and *iii)* a novel Radon transform on the Sphere. An immediate extension of this work would be to consider sliced-Wasserstein discrepancy in hyperbolic spaces, where geodesics are circular arcs as in the Poincaré disk. Beyond the generalization to other, possibly well behaved, manifolds, statistical aspects need to be examined, such as sample complexity or dependence to the hypersphere dimension. While we postulate that results comparable to the Euclidean case might be reached, the fact that the manifold is closed might bring interesting differences and justify further use of this type of discrepancies rather than their Euclidean counterparts.

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

 # A   Proofs

 ## A.1   Proof of Proposition 1

**Optimal $\alpha$.**   Let $\mu \in \mathcal{P}_2(S^1)$, $\nu = \mathrm{Unif}(S^1)$. Since $\nu$ is the uniform distribution on $S^1$, its cdf is the identity on $[0,1]$ (where we identified $S^1$ and $[0,1]$). We can extend the cdf $F$ on the real line as in [89] with the convention $F(y+1) = F(y) + 1$. Therefore, $F_\nu = \mathrm{Id}$ on $\mathbb{R}$. Moreover, we know that for all $x \in S^1$, $(F_\nu - \alpha)^{-1}(x) = F_\nu^{-1}(x + \alpha) = x + \alpha$ and

$$W_2^2(\mu, \nu) = \inf_{\alpha \in \mathbb{R}} \int_0^1 |F_\mu^{-1}(t) - (F_\nu - \alpha)^{-1}(t)|^2 \, \mathrm{d}t. \tag{25}$$

For all $\alpha \in \mathbb{R}$, let $f(\alpha) = \int_0^1 \left( F_\mu^{-1}(t) - (F_\nu - \alpha)^{-1}(t) \right)^2 \, \mathrm{d}t$. Then, we have:

$$\begin{aligned}
\forall \alpha \in \mathbb{R}, \ f(\alpha) &= \int_0^1 \left( F_\mu^{-1}(t) - t - \alpha \right)^2 \, \mathrm{d}t \\
&= \int_0^1 \left( F_\mu^{-1}(t) - t \right)^2 \, \mathrm{d}t + \alpha^2 - 2\alpha \int_0^1 (F_\mu^{-1}(t) - t) \, \mathrm{d}t \\
&= \int_0^1 \left( F_\mu^{-1}(t) - t \right)^2 \, \mathrm{d}t + \alpha^2 - 2\alpha \left( \int_0^1 x \, \mathrm{d}\mu(x) - \frac{1}{2} \right),
\end{aligned} \tag{26}$$

where we used that $(F_\mu^{-1})_\# \mathrm{Unif}([0,1]) = \mu$.

Hence, $f'(\alpha) = 0 \iff \alpha = \int_0^1 x \, \mathrm{d}\mu(x) - \frac{1}{2}$.

**Closed-form for empirical distributions.**   Let $(x_i)_{i=1}^n \in [0, 1[^n$ such that $x_1 < \cdots < x_n$ and let $\mu_n = \frac{1}{n} \sum_{i=1}^n \delta_{x_i}$ a discrete distribution.

To compute the closed-form of $W_2$ between $\mu_n$ and $\nu = \mathrm{Unif}(S^1)$, we first have that the optimal $\alpha$ is $\alpha_n = \frac{1}{n} \sum_{i=1}^n x_i - \frac{1}{2}$. Moreover, we also have:

$$\begin{aligned}
W_2^2(\mu_n, \nu) &= \int_0^1 \left( F_{\mu_n}^{-1}(t) - (t + \hat{\alpha}_n) \right)^2 \, \mathrm{d}t \\
&= \int_0^1 F_{\mu_n}^{-1}(t)^2 \, \mathrm{d}t - 2 \int_0^1 t F_{\mu_n}^{-1}(t) \mathrm{d}t - 2\hat{\alpha}_n \int_0^1 F_{\mu_n}^{-1}(t) \mathrm{d}t + \frac{1}{3} + \hat{\alpha}_n + \hat{\alpha}_n^2.
\end{aligned} \tag{27}$$

Then, by noticing that $F_{\mu_n}^{-1}(t) = x_i$ for all $t \in [F(x_i), F(x_{i+1})[$, we have

$$\int_0^1 t F_{\mu_n}^{-1}(t) \mathrm{d}t = \sum_{i=1}^n \int_{\frac{i-1}{n}}^{\frac{i}{n}} t x_i \mathrm{d}t = \frac{1}{2n^2} \sum_{i=1}^n x_i (2i - 1), \tag{28}$$

$$\int_0^1 F_\mu^{-1}(t)^2 \mathrm{d}t = \frac{1}{n} \sum_{i=1}^n x_i^2, \quad \int_0^1 F_\mu^{-1}(t) \mathrm{d}t = \frac{1}{n} \sum_{i=1}^n x_i, \tag{29}$$

and we also have:

$$\hat{\alpha}_n + \hat{\alpha}_n^2 = \frac{1}{n} \sum_{i=1}^n x_i - \frac{1}{2} + \left( \frac{1}{n} \sum_{i=1}^n x_i \right)^2 + \frac{1}{4} - \frac{1}{n} \sum_{i=1}^n x_i = \left( \frac{1}{n} \sum_{i=1}^n x_i \right)^2 - \frac{1}{4}. \tag{30}$$

Then, by plugging these results into (27), we obtain

$$\begin{aligned}
W_2^2(\mu_n, \nu) &= \frac{1}{n} \sum_{i=1}^n x_i^2 - \frac{1}{n^2} \sum_{i=1}^n (2i - 1) x_i - 2 \left( \frac{1}{n} \sum_{i=1}^n x_i \right)^2 + \frac{1}{n} \sum_{i=1}^n x_i + \frac{1}{3} + \left( \frac{1}{n} \sum_{i=1}^n x_i \right)^2 - \frac{1}{4} \\
&= \frac{1}{n} \sum_{i=1}^n x_i^2 - \left( \frac{1}{n} \sum_{i=1}^n x_i \right)^2 + \frac{1}{n^2} \sum_{i=1}^n (n + 1 - 2i) x_i + \frac{1}{12}.
\end{aligned} \tag{31}$$

## A.2 Proof of Equation (17)

Let $U \in \mathbb{V}_{d,2}$. Then the great circle generated by $U \in \mathbb{V}_{d,2}$ is defined as the intersection between $\text{span}(UU^T)$ and $S^{d-1}$. And we have the following characterization:

$$
\begin{aligned}
x \in \text{span}(UU^T) \cap S^{d-1} &\iff \exists y \in \mathbb{R}^d,\ x = UU^T y \text{ and } \|x\|_2^2 = 1 \\
&\iff \exists y \in \mathbb{R}^d,\ x = UU^T y \text{ and } \|UU^T y\|_2^2 = y^T UU^T y = \|U^T y\|_2^2 = 1 \\
&\iff \exists z \in S^1,\ x = Uz.
\end{aligned}
$$

And we deduce that

$$
\forall U \in \mathbb{V}_{d,2}, x \in S^{d-1},\ P^U(x) = \operatorname*{argmin}_{z \in S^1} d_{S^{d-1}}(x, Uz). \tag{32}
$$

## A.3 Proof of Lemma 1

Let $U \in \mathbb{V}_{d,2}$ and $x \in S^{d-1}$ such that $U^T x \neq 0$. Denote $U = (u_1 \ u_2)$, *i.e.* the 2-plane $E$ is $E = \text{span}(UU^T) = \text{span}(u_1, u_2)$ and $(u_1, u_2)$ is an orthonormal basis of $E$. Then, for all $x \in S^{d-1}$, the projection on $E$ is $p^E(x) = \langle u_1, x \rangle u_1 + \langle u_2, x \rangle u_2 = UU^T x$.

Now, let us compute the geodesic distance between $x \in S^{d-1}$ and $\frac{p^E(x)}{\|p^E(x)\|_2} \in E \cap S^{d-1}$:

$$
d_{S^{d-1}}\left( x, \frac{p^E(x)}{\|p^E(x)\|_2} \right) = \arccos\left( \langle x, \frac{p^E(x)}{\|p^E(x)\|_2} \rangle \right) = \arccos(\|p^E(x)\|_2), \tag{33}
$$

using that $x = p^E(x) + p^{E^\perp}(x)$.

Let $y \in E \cap S^{d-1}$ another point on the great circle. By the Cauchy-Schwarz inequality, we have

$$
\langle x, y \rangle = \langle p^E(x), y \rangle \leq \|p^E(x)\|_2 \|y\|_2 = \|p^E(x)\|_2. \tag{34}
$$

Therefore, using that $\arccos$ is decreasing on $(-1, 1)$,

$$
d_{S^{d-1}}(x, y) = \arccos(\langle x, y \rangle) \geq \arccos(\|p^E(x)\|_2) = d_{S^{d-1}}\left( x, \frac{p^E(x)}{\|p^E(x)\|_2} \right). \tag{35}
$$

Moreover, we have equality if and only if $y = \lambda p^E(x)$. And since $y \in S^{d-1}$, $|\lambda| = \frac{1}{\|p^E(x)\|_2}$. Using again that $\arccos$ is decreasing, we deduce that the minimum is well attained in $y = \frac{p^E(x)}{\|p^E(x)\|_2} = \frac{UU^T x}{\|UU^T x\|_2}$.

Finally, using that $\|UU^T x\|_2 = x^T UU^T UU^T x = x^T UU^T x = \|U^T x\|_2$, we deduce that

$$
P^U(x) = \frac{U^T x}{\|U^T x\|_2}. \tag{36}
$$

Finally, by noticing that the projection is unique if and only if $U^T x = 0$, and using [9, Proposition 4.2] which states that there is a unique projection for a.e. $x$, we deduce that $\{x \in S^{d-1},\ U^T x = 0\}$ is of measure null and hence, for a.e. $x \in S^{d-1}$, we have the result.

### A.4 Proof of Proposition 2

Let $f \in L^1(S^{d-1})$, $g \in C_0(S^1 \times \mathbb{V}_{d,2})$, then by Fubini's theorem,

$$
\begin{aligned}
\langle \tilde{R}f, g \rangle_{S^1 \times \mathbb{V}_{d,2}} &= \int_{V_{d,2}} \int_{S^1} \tilde{R}f(z,U)g(z,U) \, \mathrm{d}z\mathrm{d}\sigma(U) \\
&= \int_{V_{d,2}} \int_{S^1} \int_{S^{d-1}} f(x)\mathbb{1}_{\{z=P^U(x)\}}g(z,U) \, \mathrm{d}x\mathrm{d}z\mathrm{d}\sigma(U) \\
&= \int_{S^{d-1}} f(x) \int_{V_{d,2}} \int_{S^1} g(z,U)\mathbb{1}_{\{z=P^U(x)\}} \, \mathrm{d}z\mathrm{d}\sigma(U)\mathrm{d}x \\
&= \int_{S^{d-1}} f(x) \int_{V_{d,2}} g\big(P^U(x),U\big) \, \mathrm{d}\sigma(U)\mathrm{d}x \\
&= \int_{S^{d-1}} f(x)\tilde{R}^*g(x) \, \mathrm{d}x \\
&= \langle f, \tilde{R}^*g \rangle_{S^{d-1}}.
\end{aligned}
\tag{37}
$$

### A.5 Proof of Proposition 3

Let $g \in C_0(S^1 \times \mathbb{V}_{d,2})$,

$$
\begin{aligned}
\int_{\mathbb{V}_{d,2}} \int_{S^1} g(z,U) \, (\tilde{R}\mu)^U(\mathrm{d}z) \, \mathrm{d}\sigma(U) &= \int_{S^1 \times \mathbb{V}_{d,2}} g(z,U) \, \mathrm{d}(\tilde{R}\mu)(z,U) \\
&= \int_{S^{d-1}} \tilde{R}^*g(x) \, \mathrm{d}\mu(x) \\
&= \int_{S^{d-1}} \int_{\mathbb{V}_{d,2}} g(P^U(x),U) \, \mathrm{d}\sigma(U)\mathrm{d}\mu(x) \\
&= \int_{\mathbb{V}_{d,2}} \int_{S^{d-1}} g(P^U(x),U) \, \mathrm{d}\mu(x)\mathrm{d}\sigma(U) \\
&= \int_{\mathbb{V}_{d,2}} \int_{S^1} g(z,U) \, \mathrm{d}(P^U_{\#}\mu)(z)\mathrm{d}\sigma(U).
\end{aligned}
\tag{38}
$$

Hence, for $\sigma$-almost every $U \in \mathbb{V}_{d,2}$, $(\tilde{R}\mu)^U = P^U_{\#}\mu$.

### A.6 Study of the Spherical Radon transform $\tilde{R}$

In this Section, we first discuss the set of integration of the spherical Radon transform $\tilde{R}$ (19). We further show that it is related to the hemispherical Radon transform and we derive its kernel.

**Set of integration.** While the classical Radon transform integrates over hyperplanes of $\mathbb{R}^d$ and the generalized Radon transform integrates over hypersurfaces [60], the set of integration of the spherical Radon transform (19) is a half of a "big circle", *i.e.* half of the intersection between a hyperplane and $S^{d-1}$ [96]. We illustrate this on $S^2$ in Figure 7. On $S^2$, the intersection between a hyperplane and $S^2$ is a great circle.

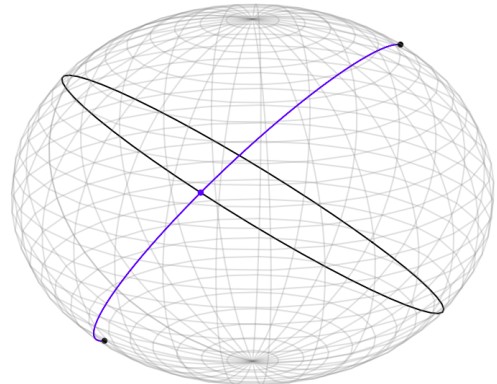

Figure 7: Set of integration of the spherical Radon transform (19). The great circle is in black and the set of integration in blue. The point $Uz \in \mathrm{span}(UU^T) \cap S^{d-1}$ is in blue.

**Proposition 6.** *Let $U \in \mathbb{V}_{d,2}$, $z \in S^1$. The set of integration of (19) is*

$$\{x \in S^{d-1}, \ P^U(x) = z\} = \{x \in F \cap S^{d-1}, \ \langle x, Uz \rangle > 0\}, \tag{39}$$

*where $F = \mathrm{span}(UU^T)^\perp \oplus \mathrm{span}(Uz)$.*

*Proof.* Let $U \in \mathbb{V}_{d,2}$, $z \in S^1$. Denote $E = \mathrm{span}(UU^T)$ the 2-plane generating the great circle, and $E^\perp$ its orthogonal complementary. Hence, $E \oplus E^\perp = \mathbb{R}^d$ and $\dim(E^\perp) = d-2$. Now, let $F = E^\perp \oplus \mathrm{span}(Uz)$. Since $Uz = UU^T Uz \in E$, we have that $\dim(F) = d-1$. Hence, $F$ is a hyperplane and $F \cap S^{d-1}$ is a "big circle" [96], *i.e.* a $(d-2)$-dimensional subsphere of $S^{d-1}$.

Now, for the first inclusion, let $x \in \{x \in S^{d-1}, \ P^U(x) = z\}$. First, we show that $x \in F \cap S^{d-1}$. By Lemma 1 and hypothesis, we know that $P^U(x) = \frac{U^T x}{\|U^T x\|_2} = z$. By denoting by $p^E$ the projection on $E$, we have:

$$p^E(x) = UU^T x = U(\|U^T x\|_2 z) = \|U^T x\|_2 Uz \in \mathrm{span}(Uz). \tag{40}$$

Hence, $x = p^E(x) + x_{E^\perp} = \|U^T x\|_2 Uz + x_{E^\perp} \in F$. Moreover, as

$$\langle x, Uz \rangle = \|U^T x\|_2 \langle Uz, Uz \rangle = \|U^T x\|_2 > 0, \tag{41}$$

we deduce that $x \in \{F \cap S^{d-1}, \ \langle x, Uz \rangle > 0\}$.

For the other inclusion, let $x \in \{F \cap S^{d-1}, \ \langle x, Uz \rangle > 0\}$. Since $x \in F$, we have $x = x_{E^\perp} + \lambda Uz$, $\lambda \in \mathbb{R}$. Hence, using Lemma 1,

$$P^U(x) = \frac{U^T x}{\|U^T x\|_2} = \frac{\lambda}{|\lambda|} \frac{z}{\|z\|_2} = \mathrm{sign}(\lambda) z. \tag{42}$$

But, we also have $\langle x, Uz \rangle = \lambda \|Uz\|_2^2 = \lambda > 0$. Therefore, $\mathrm{sign}(\lambda) = 1$ and $P^U(x) = z$.

Finally, we conclude that $\{x \in S^{d-1}, \ P^U(x) = z\} = \{x \in F \cap S^{d-1}, \ \langle x, Uz \rangle > 0\}$. $\square$

**Link with Hemispherical transform.** Since the intersection between a hyperplane and $S^{d-1}$ is isometric to $S^{d-2}$ [56], we can relate $\tilde{R}$ to the hemispherical transform $\mathcal{H}$ [96] on $S^{d-2}$. First, the hemispherical transform of a function $f \in L^1(S^{d-1})$ is defined as

$$\forall x \in S^{d-1}, \ \mathcal{H}f(x) = \int_{S^{d-1}} f(y) \mathbb{1}_{\{\langle x, y \rangle > 0\}} \mathrm{d}y. \tag{43}$$

From Proposition 6, we can write the spherical Radon transform (19) as a hemispherical transform on $S^{d-2}$.

**Proposition 7.** *Let $f \in L^1(S^{d-1})$, $U \in \mathbb{V}_{d,2}$ and $z \in S^1$, then*

$$\tilde{R}f(z, U) = \int_{S^{d-2}} \tilde{f}(x)\mathbb{1}_{\{\langle x, \tilde{U}z \rangle > 0\}}\mathrm{d}x = \mathcal{H}\tilde{f}(\tilde{U}z), \tag{44}$$

*where for all $x \in S^{d-2}$, $\tilde{f}(x) = f(O^T Jx)$ with $O$ the rotation matrix such that for all $x \in F$, $Ox \in \operatorname{span}(e_1, \ldots, e_{d-1})$ where $(e_1, \ldots, e_d)$ denotes the canonical basis, and $J = \begin{pmatrix} I_{d-1} \\ 0_{1,d-1} \end{pmatrix}$, and $\tilde{U} = J^T OU \in \mathbb{R}^{(d-1)\times 2}$.*

*Proof.* Let $f \in L^1(S^{d-1})$, $z \in S^1$, $U \in \mathbb{V}_{d,2}$, then by Proposition 6,

$$\tilde{R}f(z, U) = \int_{S^{d-1} \cap F} f(x)\mathbb{1}_{\{\langle x, Uz \rangle > 0\}}\mathrm{d}x. \tag{45}$$

$F$ is a hyperplane. Let $O \in \mathbb{R}^{d \times d}$ be the rotation such that for all $x \in F$, $Ox \in \operatorname{span}(e_1, \ldots, e_{d-1}) = \tilde{F}$ where $(e_1, \ldots, e_d)$ is the canonical basis. By applying the change of variable $Ox = y$, and since $O^{-1} = O^T$, $\det O = 1$, we obtain

$$\tilde{R}f(z, U) = \int_{O(F \cap S^{d-1})} f(O^T y)\mathbb{1}_{\{\langle O^T y, Uz \rangle > 0\}}\mathrm{d}y = \int_{\tilde{F} \cap S^{d-1}} f(O^T y)\mathbb{1}_{\{\langle y, OUz \rangle > 0\}}\mathrm{d}y. \tag{46}$$

Now, we have that $OU \in \mathbb{V}_{d,2}$ since $(OU)^T(OU) = I_2$, and since $Uz \in F$, $OUz \in \tilde{F}$. For all $y \in \tilde{F}$, we have $\langle y, e_d \rangle = y_d = 0$. Let $J = \begin{pmatrix} I_{d-1} \\ 0_{1,d-1} \end{pmatrix} \in \mathbb{R}^{d \times (d-1)}$, then for all $y \in \tilde{F} \cap S^{d-1}$, $y = J\tilde{y}$ where $\tilde{y} \in S^{d-2}$ is composed of the $d-1$ first coordinates of $y$.

Let's define, for all $\tilde{y} \in S^{d-2}$, $\tilde{f}(\tilde{y}) = f(O^T J\tilde{y})$, $\tilde{U} = J^T OU$.

Then, since $\tilde{F} \cap S^{d-1} \cong S^{d-2}$, we can write:

$$\tilde{R}f(z, U) = \int_{S^{d-2}} \tilde{f}(\tilde{y})\mathbb{1}_{\{\langle \tilde{y}, \tilde{U}z \rangle > 0\}}\mathrm{d}\tilde{y} = \mathcal{H}\tilde{f}(\tilde{U}z). \tag{47}$$

$\square$

**Kernel of $\tilde{R}$.** By exploiting the expression using the hemispherical transform in Proposition 7, we can derive its kernel in Appendix A.7.

### A.7 Proof of Proposition 4

First, we recall Lemma 2.3 of [94] on $S^{d-2}$.

**Lemma 2** (Lemma 2.3 [94]). $\ker(\mathcal{H}) = \{\mu \in \mathcal{M}_{\text{even}}(S^{d-2}), \mu(S^{d-2}) = 0\}$ *where $\mathcal{M}_{\text{even}}$ is the set of even measures,* i.e. *measures such that for all $f \in C(S^{d-2})$, $\langle \mu, f \rangle = \langle \mu, f^- \rangle$ where $f^-(x) = f(-x)$ for all $x \in S^{d-2}$.*

Let $\mu \in \mathcal{M}_{ac}(S^{d-1})$. First, we notice that the density of $\tilde{R}\mu$ w.r.t. $\lambda \otimes \sigma$ is, for all $z \in S^1$, $U \in \mathbb{V}_{d,2}$,

$$(\tilde{R}\mu)(z, U) = \int_{S^{d-1}} \mathbb{1}_{\{P^U(x)=z\}}\mathrm{d}\mu(x) = \int_{F \cap S^{d-1}} \mathbb{1}_{\{\langle x, Uz \rangle > 0\}}\mathrm{d}\mu(x). \tag{48}$$

Indeed, using Proposition 2, and Proposition 6, we have for all $g \in C_0(S^1 \times \mathbb{V}_{d,2})$,

$$\begin{aligned}
\langle \tilde{R}\mu, g \rangle_{S^1 \times \mathbb{V}_{d,2}} = \langle \mu, \tilde{R}^* g \rangle_{S^{d-1}} &= \int_{S^{d-1}} R^* g(x)\mathrm{d}\mu(x) \\
&= \int_{S^{d-1}} \int_{\mathbb{V}_{d,2}} \int_{S^1} g(z, U)\mathbb{1}_{\{z=P^U(x)\}}\mathrm{d}z\mathrm{d}\sigma(U)\mathrm{d}\mu(x) \\
&= \int_{\mathbb{V}_{d,2} \times S^1} g(z, U) \int_{S^{d-1}} \mathbb{1}_{\{z=P^U(x)\}}\mathrm{d}\mu(x) \, \mathrm{d}z\mathrm{d}\sigma(U) \\
&= \int_{\mathbb{V}_{d,2} \times S^1} g(z, U) \int_{F \cap S^{d-1}} \mathbb{1}_{\{\langle x, Uz \rangle > 0\}}\mathrm{d}\mu(x) \, \mathrm{d}z\mathrm{d}\sigma(U).
\end{aligned} \tag{49}$$

737    Hence, using Proposition 7, we can write $(\tilde{R}\mu)(z,U) = (\mathcal{H}\tilde{\mu})(\tilde{U}z)$ where $\tilde{\mu} = J^T_{\#}O_{\#}\mu$.

738    Now, let $\mu \in \ker(\tilde{R})$, then for all $z \in S^1$, $U \in \mathbb{V}_{d,2}$, $\tilde{R}\mu(z,U) = \mathcal{H}\tilde{\mu}(\tilde{U}z) = 0$ and hence
739    $\tilde{\mu} \in \ker(\mathcal{H}) = \{\tilde{\mu} \in \mathcal{M}_{\text{even}}(S^{d-2}), \ \tilde{\mu}(S^{d-2}) = 0\}$.

740    First, let's show that $\mu \in \mathcal{M}_{\text{even}}(S^{d-1})$. Let $f \in C(S^{d-1})$ and $U \in \mathbb{V}_{d,2}$, then, by using the same
741    notation as in Propositions 6 and 7, we have

$$
\begin{aligned}
\langle \mu, f \rangle_{S^{d-1}} = \int_{S^{d-1}} f(x)\mathrm{d}\mu(x) &= \int_{S^{d-1}} \int_{S^1} f(x)\mathbb{1}_{\{z = P^U(x)\}} \ \mathrm{d}z \ \mathrm{d}\mu(x) \\
&= \int_{S^1} \int_{S^{d-1}} f(x)\mathbb{1}_{\{z = P^U(x)\}}\mathrm{d}\mu(x)\mathrm{d}z \\
&= \int_{S^1} \int_{F \cap S^{d-1}} f(x)\mathbb{1}_{\{\langle x, Uz \rangle > 0\}}\mathrm{d}\mu(x)\mathrm{d}z \quad \text{by Prop. 6} \\
&= \int_{S^1} \int_{S^{d-2}} \tilde{f}(y)\mathbb{1}_{\{\langle y, \tilde{U}z \rangle > 0\}}\mathrm{d}\tilde{\mu}(y)\mathrm{d}z \\
&= \int_{S^1} \langle \mathcal{H}\tilde{\mu}, \tilde{f} \rangle_{S^{d-2}} \ \mathrm{d}z \\
&= \int_{S^1} \langle \tilde{\mu}, \mathcal{H}\tilde{f} \rangle_{S^{d-2}} \ \mathrm{d}z \\
&= \int_{S^1} \langle \tilde{\mu}, (\mathcal{H}\tilde{f})^- \rangle_{S^{d-2}} \ \mathrm{d}z \quad \text{since } \tilde{\mu} \in \mathcal{M}_{\text{even}} \\
&= \int_{S^{d-1}} f^-(x)\mathrm{d}\mu(x) = \langle \mu, f^- \rangle_{S^{d-1}},
\end{aligned}
\tag{50}
$$

742    using for the last line all the opposite transformations. Therefore, $\mu \in \mathcal{M}_{\text{even}}(S^{d-1})$.

743    Now, we need to find on which set the measure is null. We have

$$
\begin{aligned}
&\forall z \in S^1, U \in \mathbb{V}_{d,2}, \ \tilde{\mu}(S^{d-2}) = 0 \\
&\iff \forall z \in S^1, U \in \mathbb{V}_{d,2}, \ \mu(O^{-1}((J^T)^{-1}(S^{d-2}))) = \mu(F \cap S^{d-1}) = 0.
\end{aligned}
\tag{51}
$$

744    Hence, we deduce that

$$
\begin{aligned}
\ker(\tilde{R}) = \{\mu \in \mathcal{M}_{\text{even}}(S^{d-1}), \ &\forall U \in \mathbb{V}_{d,2}, \forall z \in S^1, F = \operatorname{span}(UU^T)^{\perp} \cap \operatorname{span}(Uz), \\
&\mu(F \cap S^{d-1}) = 0\}.
\end{aligned}
\tag{52}
$$

745    Moreover, we have that $\cup_{U,z} F_{U,z} \cap S^{d-1} = \{H \cap S^{d-1} \subset \mathbb{R}^d, \ \dim(H) = d-1\}$.

746    Indeed, on the one hand, let H an hyperplane, $x \in H \cap S^{d-1}$, $U \in \mathbb{V}_{d,2}$, and note $z = P^U(x)$. Then,
747    $x \in F \cap S^{d-1}$ by Proposition 6 and $H \cap S^{d-1} \subset \cup_{U,z} F_{U,z}$.

748    On the other hand, let $U \in \mathbb{V}_{d,2}$, $z \in S^1$, $F$ is a hyperplane since $\dim(F) = d-1$ and therefore
749    $F \cap S^{d-1} \subset \{H, \ \dim(H) = d-1\}$.

750    Finally, we deduce that

$$
\ker(\tilde{R}) = \{\mu \in \mathcal{M}_{\text{even}}(S^{d-1}), \ \forall H \in \mathcal{G}_{d,d-1}, \ \mu(H \cap S^{d-1})\}.
\tag{53}
$$

## A.8    Proof of Proposition 5

752    Let $p \geq 1$. First, it is straightforward to see that for all $\mu, \nu \in \mathcal{P}_p(S^{d-1})$, $SSW_p(\mu, \nu) \geq 0$,
753    $SSW_p(\mu, \nu) = SSW_p(\nu, \mu)$, $\mu = \nu \implies SSW_p(\mu, \nu) = 0$ and that we have the triangular

inequality since

$$\forall \mu, \nu, \alpha \in \mathcal{P}_p(S^{d-1}), \ SSW_p(\mu, \nu) = \Big( \int_{\mathbb{V}_{d,2}} W_p^p(P_\#^U \mu, P_\#^U \nu) \, \mathrm{d}\sigma(U) \Big)^{\frac{1}{p}}$$

$$\leq \Big( \int_{\mathbb{V}_{d,2}} \big( W_p(P_\#^U \mu, P_\#^U \alpha) + W_p(P_\#^U \alpha, P_\#^U \nu) \big)^p \, \mathrm{d}\sigma(U) \Big)^{\frac{1}{p}}$$

$$\leq \Big( \int_{\mathbb{V}_{d,2}} W_p^p(P_\#^U \mu, P_\#^U \alpha) \, \mathrm{d}\sigma(U) \Big)^{\frac{1}{p}}$$

$$+ \Big( \int_{\mathbb{V}_{d,2}} W_p^p(P_\#^U \alpha, P_\#^U \nu) \, \mathrm{d}\sigma(U) \Big)^{\frac{1}{p}}$$

$$= SSW_p(\mu, \alpha) + SSW_p(\alpha, \nu),$$

(54)

using the triangular inequality for $W_p$ and the Minkowski inequality. Therefore, it is at least a pseudo-distance.

To be a distance, we also need $SSW_p(\mu, \nu) = 0 \implies \mu = \nu$. Suppose that $SSW_p(\mu, \nu) = 0$. Since, for all $U \in \mathbb{V}_{d,2}$, $W_p^p(P_\#^U \mu, P_\#^U \nu) \geq 0$, $SSW_p^p(\mu, \nu) = 0$ implies that for $\sigma$-ae $U \in \mathbb{V}_{d,2}$, $W_p^p(P_\#^U \mu, P_\#^U \nu) = 0$ and hence $P_\#^U \mu = P_\#^U \nu$ or $(\tilde{R}\mu)^U = (\tilde{R}\nu)^U$ for $\sigma$-ae $U \in \mathbb{V}_{d,2}$ since $W_p$ is a distance on the circle. Therefore, it is a distance on the sets of injectivity of $\tilde{R}$.

### A.9 Convergence Properties

**Proposition 8.** *Let* $(\mu_k), \mu \in \mathcal{P}_p(S^{d-1})$ *such that* $\mu_k \xrightarrow[k \to \infty]{} \mu$, *then*

$$SSW_p(\mu_k, \mu) \xrightarrow[k \to \infty]{} 0.$$

(55)

*Proof.* Since the Wasserstein distance metrizes the weak convergence (Corollary 6.11 [101]), we have $P_\#^U \mu_k \xrightarrow[k \to \infty]{} P_\#^U \mu$ (by continuity) $\iff W_p^p(P_\#^U \mu_k, P_\#^U \mu) \xrightarrow[k \to \infty]{} 0$ and hence by the dominated convergence theorem, $SSW_p^p(\mu_k, \mu) \xrightarrow[k \to \infty]{} 0$. $\qquad\square$

## B  Background on the Sphere

### B.1  Uniqueness of the Projection

Here, we discuss the uniqueness of the projection $P^U$ for almost every $x$. For that, we recall some results of [9].

Let $M$ be a closed subset of a complete finite-dimensional Riemannian manifold $N$. Let $d$ be the Riemannian distance on $N$. Then, the distance from the set $M$ is defined as

$$d_M(x) = \inf_{y \in M} d(x, y).$$

(56)

The infimum is a minimum since $M$ is closed and $N$ locally compact, but the minimum might not be unique. When it is unique, let's denote the point which attains the minimum as $\pi(x)$, *i.e.* $d(x, \pi(x)) = d_M(x)$.

**Proposition 9** (Proposition 4.2 in [9])**.** *Let* $M$ *be a closed set in a complete* $m$-*dimensional Riemannian manifold* $N$. *Then, for almost every* $x$, *there exists a unique point* $\pi(x) \in M$ *that realizes the minimum of the distance from* $x$.

From this Proposition, they further deduce that the measure $\pi_\# \gamma$ is well defined on $M$ with $\gamma$ a locally absolutely continuous measure *w.r.t.* the Lebesgue measure.

In our setting, for all $U \in \mathbb{V}_{d,2}$, we want to project a measure $\mu \in \mathcal{P}(S^{d-1})$ on the great circle $\mathrm{span}(UU^T) \cap S^{-1}$. Hence, we have $N = S^{d-1}$ which is a complete finite-dimensional Riemannian manifold and $M = \mathrm{span}(UU^T) \cap S^{d-1}$ a closed set in $N$. Therefore, we can apply Proposition 9 and the push-forward measures are well defined for absolutely continuous measures.

 **B.2  Optimization on the Sphere**

785  Let $F : S^{d-1} \to \mathbb{R}$ be some functional on the sphere. Then, we can perform a gradient descent on a
786  Riemannian manifold by following the geodesics, which are the counterpart of straight lines in $\mathbb{R}^d$.
787  Hence, the gradient descent algorithm [3, 14] reads as

$$\forall k \geq 0, \ x_{k+1} = \exp_{x_k}\big( -\gamma \mathrm{grad}f(x)\big), \tag{57}$$

788  where for all $x \in S^{d-1}$, $\exp_x : T_x S^{d-1} \to S^{d-1}$ is a map from the tangent space $T_x S^{d-1} = \{v \in
789  \mathbb{R}^d, \ \langle x, v \rangle = 0\}$ to $S^{d-1}$ such that for all $v \in T_x S^{d-1}$, $\exp_x(v) = \gamma_v(1)$ with $\gamma_v$ the unique geodesic
790  starting from $x$ with speed $v$, *i.e.* $\gamma(0) = x$ and $\gamma'(0) = v$.

791  For $S^{d-1}$, the exponential map is known and is

$$\forall x \in S^{d-1}, \forall v \in T_x S^{d-1}, \ \exp_x(v) = \cos(\|v\|_2)x + \sin(\|v\|_2)\frac{v}{\|v\|_2}. \tag{58}$$

792  Moreover, the Riemannian gradient on $S^{d-1}$ is known as [3, Eq. 3.37]

$$\mathrm{grad}f(x) = \mathrm{Proj}_x(\nabla f(x)) = \nabla f(x) - \langle \nabla f(x), x \rangle x, \tag{59}$$

793  $\mathrm{Proj}_x$ denoting the orthogonal projection on $T_x S^{d-1}$.

794  For more details, we refer to [3, 17].

**B.3  Von Mises-Fisher Distribution**

796  The von Mises-Fisher (vMF) distribution is a distribution on $S^{d-1}$ characterized by a concentration
797  parameter $\kappa > 0$ and a location parameter $\mu \in S^{d-1}$ through the density

$$\forall \theta \in S^{d-1}, \ f_{\mathrm{vMF}}(\theta; \mu, \kappa) = \frac{\kappa^{d/2-1}}{(2\pi)^{d/2} I_{d/2-1}(\kappa)} \exp(\kappa \mu^T \theta), \tag{60}$$

798  where $I_\nu(\kappa) = \frac{1}{2\pi} \int_0^\pi \exp(\kappa \cos(\theta)) \cos(\nu\theta)\mathrm{d}\theta$ is the modified Bessel function of the first kind.

799  Several algorithms allow to sample from it, see *e.g.* [100, 107] for algorithms using rejection sampling
800  or [62] without rejection sampling.

801  For $d = 1$, the vMF coincides with the von Mises (vM) distribution, which has for density

$$\forall \theta \in [-\pi, \pi[, \ f_{\mathrm{vM}}(\theta; \mu, \kappa) = \frac{1}{I_0(\kappa)} \exp(\kappa \cos(\theta - \mu)), \tag{61}$$

802  with $\mu \in [0, 2\pi[$ the mean direction and $\kappa > 0$ its concentration parameter. We refer to [71, Section
803  3.5 and Chapter 9] for more details on these distributions.

804  In particular, for $\kappa = 0$, the vMF (resp. vM) distribution coincides with the uniform distribution on
805  the sphere (resp. the circle).

806  Jung [55] studied the law of the projection of a vMF on a great circle. In particular, they showed that,
807  while the vMF plays the role of the normal distributions for directional data, the projection actually
808  does not follow a von Mises distribution. More precisely, they showed the following theorem:

809  **Theorem 1** (Theorem 3.1 in [55])**.** *Let $d \geq 3$, $X \sim \mathrm{vMF}(\mu, \kappa) \in S^{d-1}$, $U \in \mathbb{V}_{d,2}$ and $T = P^U(X)$*
810  *the projection on the great circle generated by $U$. Then, the density function of $T$ is*

$$\forall t \in [-\pi, \pi[, \ f(t) = \int_0^1 f_R(r) f_{\mathrm{vM}}(t; 0, \kappa \cos(\delta) r) \ \mathrm{d}r, \tag{62}$$

811  *where $\delta$ is the deviation of the great circle (geodesic) from $\mu$ and the mixing density is*

$$\forall r \in ]0, 1[, \ f_R(r) = \frac{2}{I_\nu^*(\kappa)} I_0(\kappa \cos(\delta) r) r (1 - r^2)^{\nu-1} I_{\nu-1}^*(\kappa \sin(\delta)\sqrt{1 - r^2}), \tag{63}$$

812  *with $\nu = (d - 2)/2$ and $I_\nu^*(z) = (\frac{z}{2})^{-\nu} I_\nu(z)$ for $z > 0$, $I_\nu^*(0) = 1/\Gamma(\nu + 1)$.*

813  Hence, as noticed by Jung [55], in the particular case $\kappa = 0$, *i.e.* $X \sim \mathrm{Unif}(S^{d-1})$, then

$$f(t) = \int_0^1 f_R(r) f_{\mathrm{vM}}(t; 0, 0) \ \mathrm{d}r = f_{\mathrm{vM}(t;0,0)} \int_0^1 f_R(r)\mathrm{d}r = f_{\mathrm{vM}}(t; 0, 0), \tag{64}$$

814  and hence $T \sim \mathrm{Unif}(S^1)$.

## B.4  Normalizing Flows on the Sphere

Normalizing flows [82] are invertible transformations. There has been a recent interest in defining such transformations on manifolds, and in particular on the sphere [23, 91, 92].

Here, we implemented the Exponential map normalizing flows introduced in [92]. The transformation $T$ is

$$\forall x \in S^{d-1}, \ z = T(x) = \exp_x\big(\mathrm{Proj}_x(\nabla\phi(x))\big), \tag{65}$$

where $\phi(x) = \sum_{i=1}^{K} \frac{\alpha_i}{\beta_i} e^{\beta_i(x^T \mu_i - 1)}$, $\alpha_i \geq 0$, $\sum_i \alpha_i \leq 1$, $\mu_i \in S^{d-1}$ and $\beta_i > 0$ for all $i$. $(\alpha_i)_i$, $(\beta_i)_i$ and $(\mu_i)_i$ are the learnable parameters.

The density of $z$ can be obtained as

$$p_Z(z) = p_X(x) \det\big(E(x)^T J_T(x)^T J_T(x) E(x)\big)^{-\frac{1}{2}}, \tag{66}$$

where $J_f$ is the Jacobian in the embedded space and $E(x)$ it the matrix whose columns form an orthonormal basis of $T_x S^{d-1}$.

The common way of training normalizing flows is to use either the reverse or forward KL divergence. Here, we use them with a different loss, namely SSW.

# C  Additional Experiments

## C.1  Evolution of SSW between von Mises-Fisher distributions

The KL divergence between the von Mises-Fisher distribution and the uniform distribution has been derived analytically in [28, 110] as

$$\mathrm{KL}\big(\mathrm{vMF}(\mu,\kappa)||\mathrm{vMF}(\cdot,0)\big) = \kappa\frac{I_{d/2}(\kappa)}{I_{d/2-1}(\kappa)} + \left(\frac{d}{2}-1\right)\log\kappa - \frac{d}{2}\log(2\pi) - \log I_{d/2-1}(\kappa)$$
$$+ \frac{d}{2}\log\pi + \log 2 - \log\Gamma\left(\frac{d}{2}\right). \tag{67}$$

We plot on Figure 8 the evolution of KL and SSW *w.r.t.* $\kappa$ for different dimensions. We observe a different trend. SSW seems to get lower with the dimension contrary to KL.

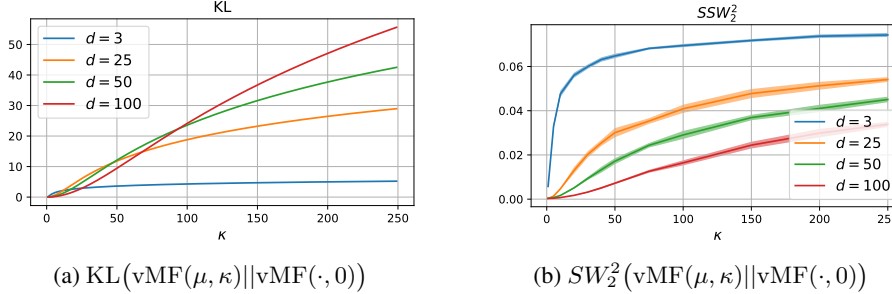

(a) $\mathrm{KL}\big(\mathrm{vMF}(\mu,\kappa)||\mathrm{vMF}(\cdot,0)\big)$      (b) $SW_2^2\big(\mathrm{vMF}(\mu,\kappa)||\mathrm{vMF}(\cdot,0)\big)$

Figure 8:  Evolution *w.r.t* $\kappa$ between $\mathrm{vMK}(\mu,\kappa)$ and $\mathrm{vMF}(\cdot,0)$.  For SW, we used 100 projections (for memory reasons for $d = 100$), and computed it for $\kappa \in \{1,5,10,20,30,40,50,75,100,150,200,250\}$, 10 times by dimension and $\kappa$, and with 500 samples of both distributions.

As a sanity check, we compare on Figure 9 the evolution of SSW between vMF distributions where we fix $\mathrm{vMF}(\mu_0, 10)$ and we rotate the first vMF along a great circle. More precisely, we plot $SW_2^2\big(\mathrm{vMF}((1,0,0,...),10), \mathrm{vMF}((\cos(\theta),\sin(\theta),0,...),10)\big)$ for $\theta \in \{\frac{k\pi}{6}\}_{k\in\{0,...,12\}}$. As expected, we obtain a bell shape which is maximal when the second vMF distribution has for location

parameter $-\mu_0$. We observe a similar behavior between $SSW_2$, $SSW_1$ and $SW_2$ with different scales.

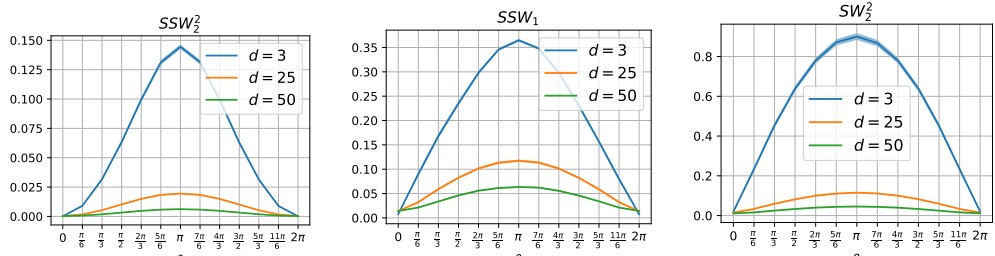

Figure 9: Evolution of $SW$ between vMF samples in $S^{d-1}$ (mean over 100 batch).

On Figure 10, we plot the evolution of SSW *w.r.t.* the number of projections for different dimensions. We observe that for around 100 projections, the variance seems to be low enough.

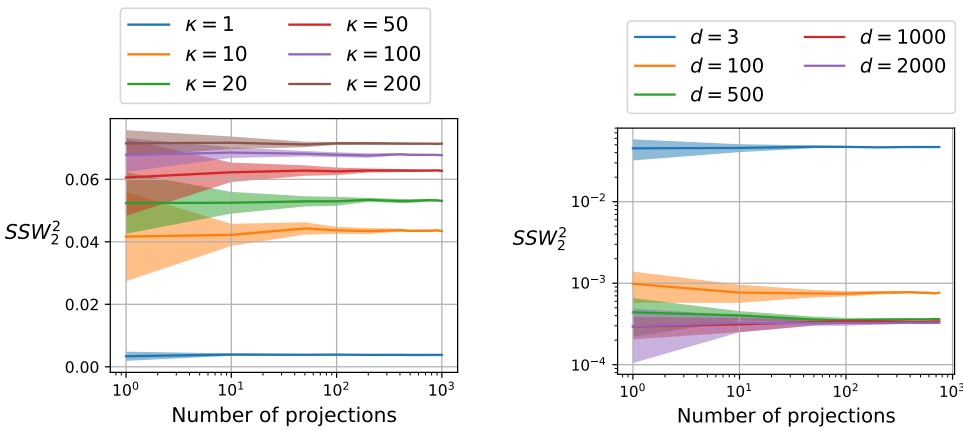

Figure 10: Influence of the number of projections. We compute $SW_2^2\big(\text{vMF}(\mu,\kappa)||\text{vMF}(\cdot,0)\big)$ 20 times, for $n = 500$ samples in dimension $d = 3$.

Nadjahi et al. [76] proved that, contrary to the Wasserstein distance, the classical sliced-Wasserstein distance has a sample complexity independent of the dimension $d$. We show empirically on Figure 11 that we expect to have similar results for SSW by plotting SSW and the Wasserstein distance (with geodesic distance) between samples of the uniform distribution on the sphere *w.r.t.* the number of samples. We observe indeed that the convergence rate of SSW is independent of the dimension.

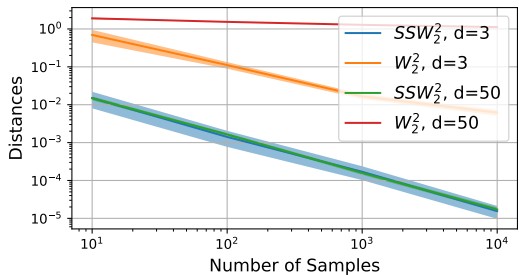

Figure 11: Spherical Sliced-Wasserstein and Wasserstein distance (with geodesic distance) between samples of the uniform distribution on the sphere. Results are averaged over 20 runs and the shaded are correponds to the standard deviation.

## C.2  Runtime Comparisons

We study here the evolution of the runtime *w.r.t.* different parameters. On Figure 12, we plot for several dimensions the runtime to compute $SSW_2$ *w.r.t.* the number of projections and the number of samples. We observe the linearity *w.r.t.* the number of projections and the quasi-linearity *w.r.t.* the number of samples.

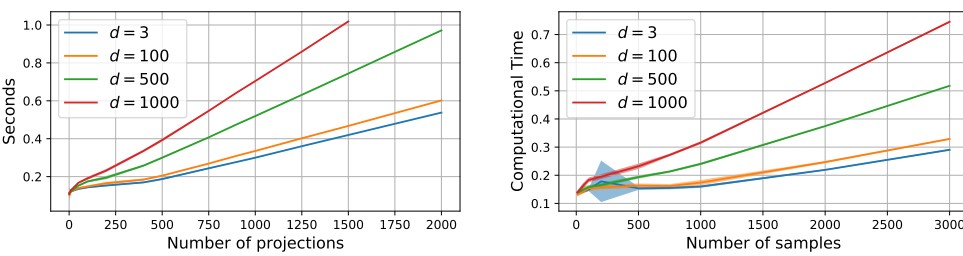

Figure 12: Computation time *w.r.t.* the number of projections or samples, taken for $\kappa = 10$ and $n = 500$ samples for the left figure, and $\kappa = 10$ and 200 projections for the right figure, and for 20 times.

## C.3  Gradient Flows

**Mixture of vMF distributions.**  For the experiment in Section 5.1, we use as target distribution of mixture of 6 vMF distributions from which we have access to samples. We refer to Appendix B.3 for background on vMF distributions.

The 6 vMF distributions have weights $1/6$, concentration parameter $\kappa = 10$ and location parameters $\mu_1 = (1, 0, 0)$, $\mu_2 = (0, 1, 0)$, $\mu_3 = (0, 0, 1)$, $\mu_4 = (-1, 0, 0)$, $\mu_5 = (0, -1, 0)$ and $\mu_6 = (0, 0, -1)$.

We use two different approximation of the distribution. First, we approximate it using the empirical distribution, *i.e.* $\hat{\mu} = \frac{1}{n} \sum_{i=1}^{n} \delta_{x_i}$ and we optimize over the particles $(x_i)_{i=1}^n$. To optimize over particles, we can either use a projected gradient descent:

$$\begin{cases} x^{(k+1)} = x^{(k)} - \gamma \nabla_{x^{(k)}} SSW_2^2(\hat{\mu}_k, \nu) \\ x^{(k+1)} = \frac{x^{(k+1)}}{\|x^{(k+1)}\|_2}, \end{cases} \tag{68}$$

or a Riemannian gradient descent on the sphere [3] (see Appendix B.2 for more details). Note that the projected gradient descent is a Riemannian gradient descent with retraction [17].

We can also use neural networks such as a multilayer perceptron (MLP). We used a MLP composed of 5 layers of 100 units with leaky relu activation functions. The output of the MLP is normalized on the sphere using a $\ell^2$ normalization. We perform a gradient descent using Adam [57] as the optimizer

with a learning rate of $10^{-4}$ for 2000 epochs. We approximate SSW with $L = 1000$ projections and a batch size of 500. The base distribution is choose as the uniform distribution on the sphere.

We report on Figure 13 a comparison of the 2 approximations where the density is estimated with a Gaussian kernel density estimator.

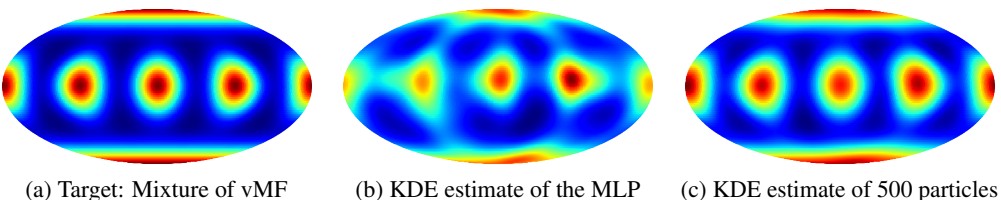

(a) Target: Mixture of vMF     (b) KDE estimate of the MLP     (c) KDE estimate of 500 particles

Figure 13: Minimization of SSW with respect to a mixture of vMF.

**vMF distribution.** A a simpler experiment, we choose a simple vMF distribution with $\kappa = 10$. We report on Figure 14 the evolution of the density approximated using a KDE, and on Figure 15 the evolution of particles.

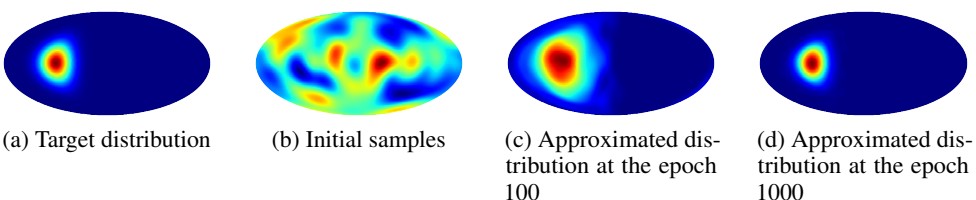

(a) Target distribution    (b) Initial samples    (c) Approximated distribution at the epoch 100    (d) Approximated distribution at the epoch 1000

Figure 14: Gradient Flows on SW with a vMF target and Mollweide projections. The distributions are approximated using KDE.

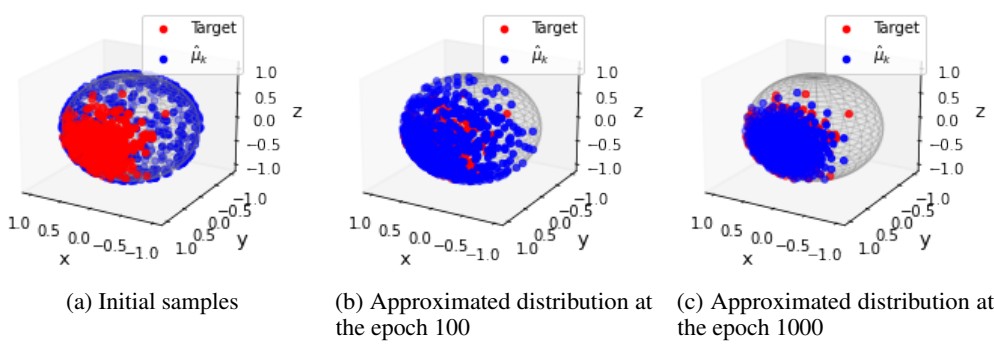

(a) Initial samples    (b) Approximated distribution at the epoch 100    (c) Approximated distribution at the epoch 1000

Figure 15: Gradient Flows on SW with a vMF target and Mollweide projections.

## C.4   Sliced-Wasserstein Variational Inference

### C.4.1   Variational Inference

In variational inference (VI) [12, 54], we have some observed data $(x_i)_{i=1}^n$ and some latent data $(z_i)_{i=1}^n$. The goal of variational inference is to approximate the posterior distribution $p(\cdot|x)$ by some distribution $q \in \mathcal{Q}$ where $\mathcal{Q}$ is a family of probabilities. The usual way of doing that is to minimize

---

**Algorithm 2** SWVI [111]

---

**Input:** $V$ a potential, $K$ the number of iterations of SWVI, $N$ the batch size, $\ell$ the number of MCMC steps
**Initialization:** Choose $q_\theta$ a sampler
**for** $k = 1$ **to** $K$ **do**
    Sample $(z_i^0)_{i=1}^N \sim q_\theta$
    Run $\ell$ MCMC steps starting from $(z_i^0)_{i=1}^N$ to get $(z_j^\ell)_{j=1}^N$
    // Denote $\hat{\mu}_0 = \frac{1}{N}\sum_{j=1}^N \delta_{z_j^0}$ and $\hat{\mu}_\ell = \frac{1}{N}\sum_{j=1}^N \delta_{z_j^\ell}$
    Compute $J = SW_2^2(\hat{\mu}_0, \hat{\mu}_\ell)$
    Backpropagate through $J$ *w.r.t.* $\theta$
    Perform a gradient step
**end for**

---

the Kullback-Leibler divergence among this family, *i.e.*

$$\min_{q \in \mathcal{Q}} \ \mathrm{KL}(q||p(\cdot|x)) = \mathbb{E}_q[\log\left(\frac{q(Z)}{p(Z|x)}\right)]. \tag{69}$$

But the KL divergence suffers from some drawbacks, as it is only a divergence (*i.e.* it does not satisfy the triangular inequality, and it is non symmetric), but it also suffers from under estimating the target distribution (or over estimating it for the reverse KL).

Yi and Liu [111] propose to use an optimal transport distance instead, namely the SW distance which gives the sliced-Wasserstein variational inference method. Basically, given some unnormalized probability $p(\cdot|x)$ that we want to approximate with some variational distribution $q_\phi$, we can first apply a MCMC algorithm and then learn $q_\phi$ using a gradient descent on SW with the target being the empirical distributions of the samples given by the MCMC. But running long MCMC chain is time consuming and it might be difficult to diagnose burn-in period. Therefore, they propose to only run at each iteration some number of steps $t$ of MCMC chain, and then learn by gradient descent the variational distribution. Therefore, the variational distribution is guided at each step by the MCMC samples toward the stationary distribution which is the target. This is called an amortized sampler (see Problem 1 in [103]). We sum up the procedure in Algorithm 2.

We propose here to substitute $SW$ by $SSW$ in order to perform SSWVI on the sphere. To do that, we first need a MCMC method on the sphere.

### C.4.2 MCMC on the Sphere

Several MCMC methods on the sphere have been proposed. For example, Hamiltonian Monte-Carlo (HMC) methods were proposed in [18, 63, 68], and Riemannian Langevin algorithms were proposed in [65, 105].

In our experiments, we use the Geodesic Langevin algorithm (GLA) introduced by Wang et al. [105]. This algorithm is a natural generalization of the Unadjusted Langevin Algorithm (ULA) and it consists at simply following the geodesics of the regular ULA step, *i.e.*

$$\forall k > 0, \ x_{k+1} = \exp_{x_k}\left(\mathrm{Proj}_{x_k}(-\gamma\nabla V(x_k) + \sqrt{2\gamma}Z)\right), \ Z \sim \mathcal{N}(0, I), \tag{70}$$

where for the sphere,

$$\forall x \in S^{d-1}, \forall v \in T_x S^{d-1}, \ \exp_x(v) = x\cos(\|v\|) + \frac{v}{\|v\|}\sin(\|v\|), \tag{71}$$

$\mathrm{Proj}_x$ is the projection on the tangent space $T_x S^{d-1} = \{v \in \mathbb{R}^d, \ \langle x, v \rangle = 0\}$ (which is the orthogonal space) and is defined as

$$\mathrm{Proj}_x(v) = v - \langle x, v \rangle x. \tag{72}$$

For more details, we refer to [3].

We use GLA here for simplicity and as a proof of concept. But note that GLA, as ULA, is biased and therefore the distribution learned will not be the exact true stationary distribution. However, a Metropolis-Hastings step at each iteration could be used to enforce the reversibility *w.r.t.* the target distribution or we could use other MCMC with more appealing convergence properties (see *e.g.* [68]).

### C.4.3 Applications

**Target: Power spherical distribution.** First, as a simple example on $S^2$, we use the power spherical distribution introduced by De Cao and Aziz [29]. This distribution has the advantage over the vMF distribution to allow for the direct use of the reparameterization trick since it does not require rejection sampling. The pdf is obtained as,

$$\forall x \in S^{d-1}, \; p_X(x; \mu, \kappa) \propto (1 + \mu^T x)^\kappa \tag{73}$$

with $\mu \in S^{d-1}$ and $\kappa > 0$. We can sample from drawing first $Z \sim \text{Beta}(\frac{d-1}{2} + \kappa, \frac{d-1}{2})$, $v \sim \text{Unif}(S^{d-2})$, then constructing $T = 2Z - 1$ and $Y = [T, v^T\sqrt{1 - T^2}]^T$. Finally, apply a Householder reflection about $\mu$ to $Y$. All the operations are well differentiable and allow to apply the reparametrization trick. For the algorithm, see Algorithm 1 in [29]. Hence, in this case, if we denote $g_\theta$ the map which takes samples from a uniform distribution on $S^{d-2}$ and from a Beta distribution as input and outputs samples of power spherical distribution with parameters $\theta = (\kappa, \mu)$, we can use it as the sampler. We test the algorithm with a target being a power spherical distribution of parameter $\mu = (0, 1, 0)$ and $\kappa = 10$, starting from $\mu = (1, 1, 1)$ and $\kappa = 0.1$. Performing 2000 optimization steps with a gradient descent (Riemannian gradient descent on $\mu$ to stay on the sphere), and 20 steps of the GLA algorithm, we are getting close enough to the true distribution as we can see on Figure 16.

For the hyperparameters, we used a step size of $10^{-3}$ for GLA, 1000 projections to approximate SSW, a Riemannian gradient descent on the sphere [3] to learn the location parameter $\mu$ with a learning rate of 2, and a learning of 200 for $\kappa$. We performed $K = 2000$ steps and used $N = 500$ particles.

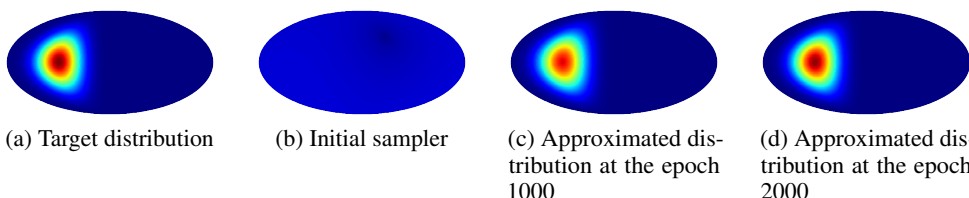

(a) Target distribution     (b) Initial sampler     (c) Approximated distribution at the epoch 1000     (d) Approximated distribution at the epoch 2000

Figure 16: SWVI on Power Spherical Distributions with Mollweide projections.

**Target: mixture of vMFs.** In Section 5.1, we perform amortized variational inference with a mixture of vMF distributions as target. For this, we train exponential map normalizing flows (see [92] and Appendix B.4). Moreover, we use the same target as Rezende et al. [92], *i.e.* the target $\nu$ has a density $p(x) \propto \sum_{k=1}^4 e^{10x^T T_{s\to e}(\mu_k)}$ with $\mu_1 = (0.7, 1.5)$, $\mu_2 = (-1, 1)$, $\mu_3 = (0.6, 0.5)$ and $\mu_4 = (-0.7, 4)$. These are spherical coordinates which are be converted to euclidean using $T_{s\to e}(\theta, \phi) = (\sin\phi\cos\theta, \sin\phi\sin\theta, \cos\phi)$.

The exponential map normalizing flow is composed of $N = 6$ blocks with $K = 5$ components. We run the algorithm for 10000 iterations, with at each iteration 20 steps of GLA with $\gamma = 10^{-1}$ as learning rate, and one step of backpropagation through SSW using the Adam [57] optimizer with a learning rate of $10^{-3}$.

We report on Figure 4 the Mollweide projection of the learned density. Since we learn to samples from a noise distribution, here the uniform distribution on the sphere, we do not have directly access to the density and we report a kernel density estimate with a Gaussian kernel using the implementation of Scipy [102].

We also report in Figure 5 the effective sample size (ESS) [33, 69] over the iterations. The ESS is estimated by [92]

$$\text{ESS} = \frac{\text{Var}_{Unif}(e^{-\beta u(X)})}{\text{Var}_q\left(\frac{e^{-\beta u(X)}}{q_\eta(X)}\right)} \approx \frac{\left(\sum_{s=1}^{S} w_s\right)^2}{\sum_{s=1}^{S} w_s^2}, \tag{74}$$

where $w_s = e^{-\beta u(x_s)/q_\eta(x_s)}$. The ESS is reported as a percentage of the sample size. Higher ESS indicates that the flow matches the target better [92].

## C.5 Sliced-Wasserstein Autoencoder

We recall that in the WAE framework, we want to minimize

$$\mathcal{L}(f, g) = \int c(x, g(f(x))) \mathrm{d}\mu(x) + \lambda D(f_\# \mu, p_Z), \tag{75}$$

where $f$ is an encoder, $g$ a decoder, $p_Z$ a prior distribution, $c$ some cost function and $D$ is a divergence in the latent space. Several $D$ were proposed. For example, Tolstikhin et al. [99] proposed to use the MMD, Kolouri et al. [59] used the SW distance, Patrini et al. [84] used the Sinkhorn divergence, Kolouri et al. [60] used the generalized SW distance. Here, we use $D = SSW_2^2$.

**Architecture and procedure.** For the encoder $f$ and the decoder $g$, we use the same architecture as Kolouri et al. [59].

For both the encoder and the decoder architecture, we use fully convolutional architectures with 3x3 convolutional filters. More precisely, the architecture of the encoder is

$$\begin{aligned}
x \in \mathbb{R}^{28 \times 28} &\to \text{Conv2d}_{16} \to \text{LeakyReLU}_{0.2} \\
&\to \text{Conv2d}_{16} \to \text{LeakyReLU}_{0.2} \to \text{AvgPool}_2 \\
&\to \text{Conv2d}_{32} \to \text{LeakyReLU}_{0.2} \\
&\to \text{Conv2d}_{32} \to \text{LeakyReLU}_{0.2} \to \text{AvgPool}_2 \\
&\to \text{Conv2d}_{64} \to \text{LeakyReLU}_{0.2} \\
&\to \text{Conv2d}_{64} \to \text{LeakyReLU}_{0.2} \to \text{AvgPool}_2 \\
&\to \text{Flatten} \to \text{FC}_{128} \to \text{ReLU} \\
&\to \text{FC}_{d_Z} \to \ell^2 \text{ normalization}
\end{aligned}$$

where $d_Z$ is the dimension of the latent space (either 11 for $S^{10}$ or 3 for $S^2$).

The architecture of the decoder is

$$\begin{aligned}
z \in \mathbb{R}^{d_Z} &\to \text{FC}_{128} \to \text{FC}_{1024} \to \text{ReLU} \\
&\to \text{Reshape(64x4x4)} \to \text{Upsample}_2 \to \text{Conv}_{64} \to \text{LeakyReLU}_{0.2} \\
&\to \text{Conv}_{64} \to \text{LeakyReLU}_{0.2} \\
&\to \text{Upsample}_2 \to \text{Conv}_{64} \to \text{LeakyReLU}_{0.2} \\
&\to \text{Conv}_{32} \to \text{LeakyReLU}_{0.2} \\
&\to \text{Upsample}_2 \to \text{Conv}_{32} \to \text{LeakyReLU}_{0.2} \\
&\to \text{Conv}_1 \to \text{Sigmoid}
\end{aligned}$$

To compare the different autoencoders, we used as the reconstruction loss the binary cross entropy, $\lambda = 10$, Adam [57] as optimizer with a learning rate of $10^{-3}$ and Pytorch's default momentum parameters for 800 epochs with batch of size $n = 500$. Moreover, when using SW type of distance, we approximated it with $L = 1000$ projections.

We report in Table 1 the FID obtained using 10000 samples and we report the mean over 5 trainings.

For SSW, we used the formulation using the uniform distribution (12). To compute SW, we used the POT library [39]. To compute the Sinkhorn divergence, we used the GeomLoss package [37].

963 **Additional experiments.** We report on Figure 17 samples obtained with SSW for a uniform prior
964 on $S^{10}$.

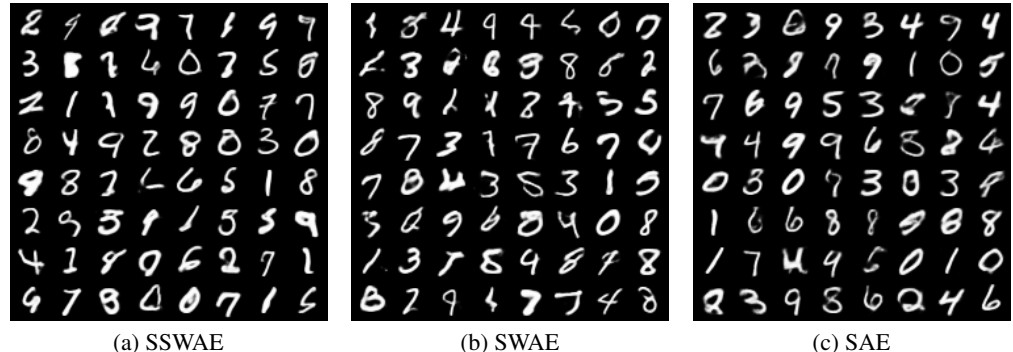

| (a) SSWAE | (b) SWAE | (c) SAE |

Figure 17: Samples generated with Sliced-Wasserstein Autoencoders with a uniform prior on $S^{10}$.

965 On Figure 18, we add the evolution over epochs of the Wasserstein distance between generated
966 images and samples from the test set.

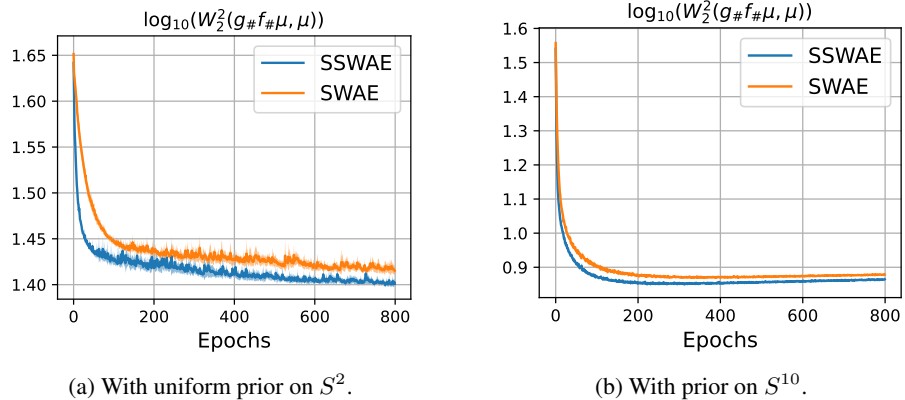

(a) With uniform prior on $S^2$.  (b) With prior on $S^{10}$.

Figure 18: Comparison of the evolution of the Wasserstein distance over epochs between SWAE and
SSWAE on MNIST (averaged over 5 trainings).

## C.6 Self-supervised learning

968 We conduct experiments using SSW to
969 prevent collapsing representations in con-
970 trastive self-supervised learning (SSL)
971 models. Such contrastive losses on the hy-
972 persphere have exhibited great representa-
973 tive capacity [20, 21, 108] on unlabelled
974 datasets by learning robust image represen-
975 tations invariantly to augmentations. As
976 proposed in [104], the contrastive objec-
977 tive can be decomposed into an alignment
978 loss which forces positive representations
979 coming from the same image to be similar

Table 2: Linear evaluation on CIFAR10. The features
are taken either on the encoder output or directly on the
sphere $S^2$.

| Method | Encoder output | $S^2$ |
|---|---|---|
| Supervised | 82.26 | 81.43 |
| Chen et al. [21] | 66.55 | 59.09 |
| Wang and Isola [104] | 60.53 | 55.86 |
| SW-SSL, $\lambda = 1, L = 10$ | 62.65 | 57.77 |
| SW-SSL, $\lambda = 1, L = 3$ | 62.46 | 57.64 |
| SSW-SSL, $\lambda = 20, L = 10$ | 64.89 | 58.91 |
| SSW-SSL, $\lambda = 20, L = 3$ | 63.75 | 59.75 |

980 and a uniformity loss which preserves maximal information of the feature distribution and hence
981 avoids collapsing representations. Without the uniformity loss, the representations tend to converge

towards a constant representation which yields the best alignment loss possible but also contains no information about original images. Wang and Isola [104] propose to enforce uniformicity by leveraging the Gaussian potential kernel which is bound to the uniform distribution on the sphere. This formulation is also related to the denominator of the contrastive loss as specified in Chen et al. [21]. We propose to replace the Gaussian kernel uniformity loss with SSW for which the complexity is more linear *w.r.t.* the number of batch samples. A simple choice of the alignment loss is to minimize the mean squared euclidean distance between pairs of different augmented versions of the same image. A self-supervised learning network is pre-trained using this alignment loss added with an uniformity term. Our overall self-supervised loss can be defined as:

$$\mathcal{L}_{\text{SSW-SSL}} = \underbrace{\frac{1}{n}\sum_{i=1}^{n}\|z_i^A - z_i^B\|_2^2}_{\text{Alignment loss}} + \frac{\lambda}{2}\big(\underbrace{SSW_2^2(z^A,\nu) + SSW_2^2(z^B,\nu)}_{\text{Uniformity loss}}\big), \tag{76}$$

where $z^A, z^B \in \mathbb{R}^{n\times d}$ are the representations from the network projected on the hypersphere of two augmented versions of the same images, $\nu = \text{Unif}(S^{d-1})$ is the uniform distribution on the hypersphere and $\lambda > 0$ is used to balance the two terms.

We pretrain a ResNet18 [47] model on the CIFAR10 [61] data with projections projected onto the sphere $S^2$. This feature dimension allow us to visualize the entire validation set of CIFAR10 and its distribution on the sphere. The visualization of the projections on $S^2$ are visible on Figure 19. We then evaluate the performance of each contrastive objective by fitting a linear classifier on top of the output of the layer before the projection on the sphere on the training dataset as is common for SSL methods. For comparison, we also report the results when the features are taken directly on the sphere. As a baseline, we also train a predictive supervised encoder by training jointly the linear classifier and the image encoder in a supervised manner using cross entropy.

We use a ResNet18 [47] encoder which outputs 1024 features that are then projected onto the sphere $S^2$ using a last fully connected layer followed by a $\ell^2$ normalization. We pretrain the model for 200 epochs using minibatch stochastic gradient descent (SGD) with a momentum of 0.9, a weight decay of 0.001 and an initial learning rate of 0.05. We use a batch size of 512 samples. The images are augmented using a standard set of random augmentations for SSL: random crops, horizontal flipping, color jittering and gray scale transformation as done in Wang and Isola [104]. For the trade-off parameter $\lambda$, we $\lambda = 20$ for SSW and $\lambda = 1$ for SW.

To evaluate the performance of representations, we use the common linear evaluation protocol where a linear classifier is fitted on top of the pre-trained representations and the best validation accuracy is reported. The linear classifiers are trained for 100 epochs using the Adam [57] optimizer with a learning rate of 0.001 with a decay of 0.2 at epoch 60 and 80. We compare our methods with two other contrastive objectives, Chen et al. [21] with the normalized temperature-scaled cross-entropy (NT-Xent) loss and Wang and Isola [104] which proposes to decompose the objective in two distinct terms $\mathcal{L}_{\text{align}}$ and $\mathcal{L}_{\text{uniform}}$. We recall the respective uniformity loss of each method in Table 3. As one can see in Table 2, our method achieves here comparable performances to two state-of-the-art approaches, yet slightly under-performing compared to [21]. We suspect that a finer validation of the balancing parameter $\lambda$ is needed. Especially since the representations on Figure 19b are not completely uniformly distributed around the sphere after pre-training compared to other contrastive methods. Nevertheless, these preliminary results show that SSW-SSL is a promising contrastive learning approach without explicit distances between negative samples, especially compared to SW on the sphere. To this end, further works should be devoted to finding a good balance between the alignment and uniformity objectives.

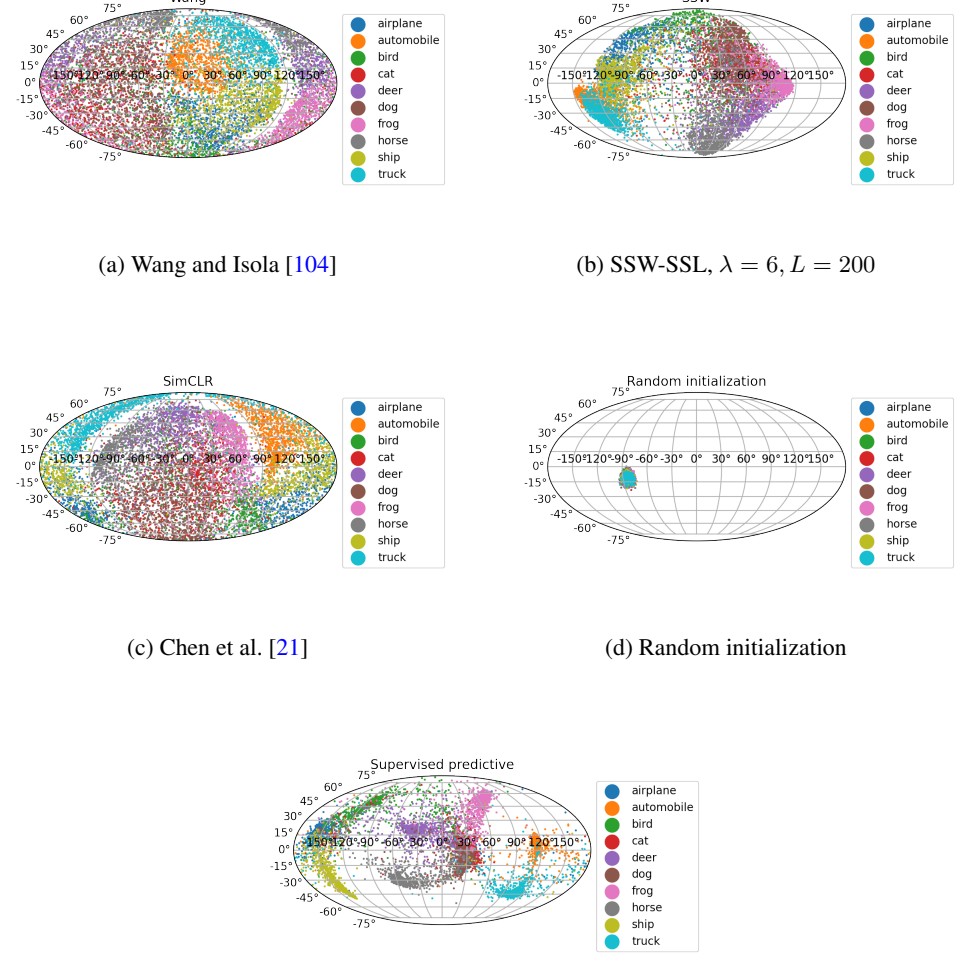

(a) Wang and Isola [104]

(b) SSW-SSL, $\lambda = 6, L = 200$

(c) Chen et al. [21]

(d) Random initialization

(e) Supervised prediction

Figure 19: The CIFAR10 validation set on $S^2$ after pre-training.

Table 3: Comparison of contrastive methods and their respective uniformity objective where $z^A, z^B \in \mathbb{R}^{n \times d}$ are representations from two augmented versions of the same set of images and $\nu = \text{Unif}(S^{d-1})$ is the uniform distribution on the hypersphere.

| Method | $\mathcal{L}_{\text{uniform}}(z^A) + \mathcal{L}_{\text{uniform}}(z^B)$ | Complexity |
|---|---|---|
| Chen et al. [21] | $\frac{1}{2n} \sum_{i=1}^{n} \log \sum_{j \neq i} \exp(\frac{\langle \hat{z}_i, \hat{z}_j \rangle}{\tau}), \hat{z} = \text{cat}(z^A, z^B)$ | $O(n^2 d)$ |
| Wang and Isola [104] | $\sum_{z \in \{z^A, z^B\}} \log \frac{2}{n(n-1)} \sum_{i>j} \exp(-t\|z_i - z_j\|_2^2)$ | $O(n^2 d)$ |
| SSW-SSL (Ours) | $\frac{1}{2}(SSW_2^2(z^A, \nu) + SSW_2^2(z^B, \nu))$ | $O(Ln(d + \log n))$ |

