# OpenReview forum: "Spherical Sliced-Wasserstein"
_NeurIPS.cc/2022/Conference — NeurIPS 2022 Submitted_

### Official Review · Reviewer_q6N2 · 2022-06-14

**Rating:** 7
**Confidence:** 3
**Soundness:** 4 excellent
**Presentation:** 3 good
**Contribution:** 3 good

**Summary:**

The paper introduces sliced Wasserstein discrepancies defined on the hypersphere $S^{d-1}$.

From a computational perspective, this corresponds to computing an expected Wasserstein distance on great circles.

From a theoretical perspective, the authors introduce a new spherical Radon transform that allows to formally define the proposed discrepancy. Further, this formal treatment allows to show that the proposed discrepancy is at least a pseudo-distance on the set of distributions on $S^{d-1}$ with finite $L_p$-norm.

Finally, the authors conclude with various experiments ranging from simple gradient-flow examples to more sophisticated variational inference or auto encoder experiments.

**Questions:**

* Was the improved separation of the latent space for SSAWE consistent among seeds? (Figure 6)
* What is the measure of variability in Figure 5? Standard deviation? Standard error? And how many intervals are visualized? Two times?

**Limitations:**

In general, the authors do not make exaggerated claims about applicability or potential of the method. In my opinion, only the statement in line 289/290 regarding improvement in the context of VI may be a bit exaggerated/unclear, as it is not indicated which variability measure is visualized in Figure 5 and further the measures overlap. Consequently, the authors could simply say that SSWVI performs just as well as SWVI.

Finally, a common problem of sliced Wasserstein distances is that the required number of projections potentially grow exponentially with dimension (as e.g. argued for in the "Generalized Sliced Wasserstein Distances" paper by Kolouri et al.). The authors could mention this shortcoming and further link it to Figure 8b) in the appendix, as the reduction of $SSW_2^2$ distance with increasing dimension may indeed by explained by the aforementioned dimensionality issue of any sliced Wasserstein distance.

**Strengths And Weaknesses:**

The paper is overall well-written and easy to follow. The theoretical claims sound reasonable to me, although I am not a mathematician and not particularly specialized in the theory of Radon transforms. Finally, the experiments are insightful and demonstrate that the discrepancy measure behaves as expected.

While the experiments do not show a strong performance increase over "naively" using the standard sliced Wasserstein distance, I still believe that the paper makes an important contribution as the introduced discrepancy is clearly more suited to the space $S^{d-1}$. Further, Figure 6 seems to show an improved class separation in the auto encoder experiment.

I see the following points for further improving the paper:
 * The authors could more clearly highlight the reason why their paper is not just a particular instance of the generalized sliced Wasserstein distance (GSW) by Kolouri et al. If I understand correctly, GSW treats the "projected particles" as members of the real line, and hence cannot take the particular geometry of a circle into account. Or would it be possible to define the proposed discrepancy as a particular instance of a GSW distance?
 * Following up on the previous point, the authors could additionally compare to GSW in their experiments. If the proposed method can indeed not be defined as a GSW instance, the GSW approach would I guess correspond to an intermediate version between regular SW and SSW, which uses the projection onto the great circles but does not respect the circular nature of the projected space.

---

> ### Author Response · Authors · 2022-08-02
> **Response to Reviewer q6N2**
>
>
> We thank the reviewer for their comments and their kind words about this work. We answer their questions and concerns below.
>
> **Link with GSW.**
>
> SSW is not a particular instance of GSW for different reasons. First of all, GSW is based on using a defining function $g:\mathcal{X}\times (\mathbb{R}^n\setminus\{0\})\to\mathbb{R}$ where $\mathcal{X}\subset\mathbb{R}^d$ and which needs to satisfy different properties. By analogy, our defining function would be defined on $\mathcal{X}\times\mathbb{V}_{d,2}$ as $g(x,U)=P^U(x)$ (with value in $S^1$).
>
> But, the defining function needs to satisfy 4 properties. One of these properties is homogeneity, i.e. that for all $\lambda\in\mathbb{R}$ , $g(x,\lambda\theta) = \lambda g(x,\theta)$. In our case, if we relax $U\in\mathbb{V}_{d,2}$ by $U\in\mathbb{R}^{d\times 2}$, then we have for all $\lambda\in\mathbb{R}$, $g(x,\lambda U) = P^{\lambda U}(x) = \lambda \frac{U^T x}{\|\lambda U^T x\|_2} = P^U(x)$ (using the closed-form of Lemma 1). Hence, it is not homogeneous but scale invariant (which makes sense since we are restricted on the sphere). Therefore, this Radon transform is not a generalized Radon transform and SSW does not enter in the framework of GSW.
>
> Moreover, another difference is indeed that the projection is on $S^1$ while the projection of GSW in on $\mathbb{R}$. However, it would be interesting to see if we can define analogous generalized Radon transform on manifolds by using only objects well defined on those.
>
> **Comparison with GSW.** We compare with GSW for autoencoders and find that SSWAE performs slightly better than GSWAE. We also compared on the variational inference experiment and found similar performances between SW, SSW and Circular GSW.
>
> **Was the improved separation of the latent space for SSWAE consistent among seed?** The improved separation over the latent space for a uniform prior was observed consistently by running several time the experiments with different seeds (see e.g. https://ibb.co/k0BFx3D for 4 different latent spaces obtained with different seeds). We added this observation in the revised version of the paper.
>
> **What is the measure of variability in Figure 5? Standard deviation? Standard error?** Thank you for pointing this out. As a matter of fact, we did not specify the measure of variability and will add it in the camera ready version of the work. We plot in Figure 5 a 95% confidence interval.
>
> **Claim about performances for Variational inference.** We will change it in “SSWVI performs as well as SWVI.”
>
> **Dimension issue.** We thank the reviewer for this suggestion and will add it to the next version of the paper. Moreover, we note that we could also define the same type of variants as SW to alleviate dimensional issues related to projections (e.g. max-SSW…).

---

### Official Review · Reviewer_yBhy · 2022-06-20

**Rating:** 5
**Confidence:** 4
**Soundness:** 3 good
**Presentation:** 3 good
**Contribution:** 3 good

**Summary:**

The paper proposes spherical sliced Wasserstein discrepancy, a variant of sliced Wasserstein on the sphere. To construct the discrepancy, the authors develop a new variant of Radon Transform which is spherical Radon Transform, and utilize the closed-form solutions of the Wasserstein distance on the circle. In more detail, the spherical sliced Wasserstein is defined as the Wasserstein distance between geodesic projected measures on the great circles measured by the uniform distribution over all the projections or Stiefel manifold. On the application side, the authors apply the new discrepancy to hyperspherical auto-encoders and density estimation on the sphere.

**Questions:**

* One benefit of sliced Wasserstein is a good sample complexity which escapes the curse of dimensionality of the Wasserstein distance. Is this still hold for spherical sliced Wasserstein?
* It seems that both spherical sliced Wasserstein and sliced Wasserstein need to use empirical samples when dealing with continuous measures in hyperspherical autoencoder. This can be seen as the usage of mini-batch OT [1,2,3,4,5]. Can we derive some special cases of spherical sliced Wasserstein where the Wasserstein distance between projected continuous measures has closed-form e.g, von Mises Fisher distribution?
* One paper might be related [6].

I will raise my score if the authors can add additional experiments on real datasets e.g., hyperspherical autoencoder on CIFAR10, CelebA, estimating density on Tiff dataset [7], and so on.

[1] "Learning with minibatch Wasserstein : asymptotic and gradient properties"
[2] "Minibatch optimal transport distances; analysis and applications"
[3] "Improving Mini-batch Optimal Transport via Partial Transportation"
[4] "On Transportation of Mini-batches: A Hierarchical Approach"
[5] "Unbalanced minibatch Optimal Transport; applications to Domain Adaptation"
[6] "Improving Relational Regularized Autoencoders with Spherical Sliced Fused Gromov Wasserstein"
[7] "https://sedac.ciesin.columbia.edu/data/set/gpw-v4-population-density-adjusted-to-2015-unwpp-country-totals-rev11/data-download#"

**Limitations:**

The paper develops a fundamental tool hence there is no foreseen negative societal impact.

**Strengths And Weaknesses:**

# Strengths
## Originality
* The paper proposes the first variant of sliced Wasserstein on the hypersphere.
* Spherical Radon Transform is new.

## Quality
* The authors derive the metricity of spherical sliced Wasserstein, the connection of spherical Radon Transform and geodesic projection, and the kernel of spherical Radon Transform.

## Clarity
* The paper is well-written and easy to follow.

## Significance
* The contribution of the paper is significant for estimating densities and modeling on the hypersphere and for the community of sliced Wasserstein and optimal transport.
* Spherical sliced Wasserstein is better than sliced Wasserstein on variational inference task on the sphere.

# Weaknesses
* The injectivity of spherical Radon Transform has not been established yet hence spherical sliced Wasserstein is not a metric.
* The experiments are not intensive. The only experiment on real data in the main paper is hyper-spherical AE on MNIST. However, the improvement is not very significant compared to the conventional sliced Wasserstein. Other datasets such as CIFAR10 and CelebA should be considered.
* The experiments on self-supervised learning should be moved to the main text.

---

> ### Author Response · Authors · 2022-08-02
> **Response to Reviewer yBhy (Part 1)**
>
>
> We thank the reviewer for their thorough review. We address their concern and questions below.
>
> **SSW is not a metric.** We would like to emphasize that we do not know yet whether or not SSW is a metric. We conjecture that it is one, but we would need an expertise in measure theory to find out whether the set of injectivity of the spherical Radon transform is null or not. Moreover, even if SSW is not a metric, it is still a pseudo distance, and we showed in our experiments that it gives performances comparable to SW, which is promising as it is defined only by using objects well defined on manifolds.
>
> **The experiments are not intensive.**
>
> We added in the revised version of the paper an experiment on the Fashion MNIST dataset which is slightly more complicated than MNIST. We still choose a uniform prior on $S^{10}$. We obtain  on this dataset better results with SSWAE compared to other methods. We report in the next table these results. We also provide here results on CIFAR10 obtained for $\lambda=0.1$, a latent space of dimension 64, 100 epochs, and averaged over 5 runs. We see that the results are pretty closed and it is hence difficult to conclude on which is the better metric for this task.
>
> | Method \ Dataset | Fashion | CIFAR10 |
> |------------------|---------|---------|
> | SSWAE | **43.94 $\pm$ 0.81**| 98.57 $\pm$ 0.35 |
> | SWAE | 44.78 $\pm$ 1.07 | **98.5 $\pm$ 0.45** |
> | WAE-MMD-IMQ| 68.51 $\pm$ 2.76 | 100.14 $\pm$ 0.67 |
> | WAE-MMD-RBF | 70.58 $\pm$ 1.75 | 100.27 $\pm$ 0.74 |
> | SAE | 56.75 $\pm$ 1.7 | 99.34 $\pm$ 0.96 |
> | Circular GSWAE | 44.65 $\pm$ 1.2 | - |
>
> We did not have time to run the full experiments on other datasets such as CelebA, given it requires to find the right learning parameters and regularization strengths. Nevertheless we will include them in the camera ready version of the paper.
>
>
> About the tiff dataset suggested by the reviewer, or other real dataset on earth. Thank you for the suggestion, as those data are naturally embed on $S^2$. Yet, how to define a meaningfull learning task from them, as well as being able to compare quantitatively competing methods, is not clear for us, and we think the corresponding illustrations would have been redundant with other results presented in the paper. Therefore, we chose to focus on tackling higher dimensional experiments such as learning the latent space of autoencoders or the self-supervised learning experiment.
>
>
>
> **The experiments on self-supervised learning should be moved to the main text.** If accepted, we will add the SSL experiment in the main text.
>
> **Sample Complexity.** We do not have a theoretical result about the sample complexity of SSW. However, we conjecture that the same type of results as for SW holds. We will add in Appendix a plot of the empirical sample complexity of W with geodesic distance and SSW (see Figure 11 of the revised version or https://ibb.co/Y05W9yS). We observe that, contrary to W and similarly to the classical SW distance, the sample complexity of SSW seems not to depend on the dimension.

---

> > ### Author Response · Authors · 2022-08-02
> > **Response to Reviewer yBhy (Part 2)**
> >
> > **Dealing with continuous measure.**
> >
> > Finding a closed-form for the Wasserstein distance between continuous measures is a hard problem in practice only solved to our knowledge for Gaussians and elliptical distributions. Distributions on the sphere are in general more complicated and it is therefore still an open question whether or not we can derive closed-forms for the Wasserstein distance between them, even on the circle. We do not know any results on von Mises distributions for example. Moreover, deriving a closed-form between projected measures is even more difficult since the projected measures can follow a different distribution. For example, the projection of a von Mises-Fisher distribution on a great circle does not follow a von Mises distribution in practice [1], but an infinite mixture of von Mises distributions. The Wasserstein distance between other generalizations of Gaussians on the sphere still need to be studied (e.g. the Riemannian normal distribution [2] or the wrapped normal distribution [3]) or other distributions such as the power spherical distribution [4].
> >
> > Note however that we derived in Proposition 1 a closed-form for the Wasserstein distance on the circle between an arbitrary distribution and the uniform measure.
> >
> > **Additional reference.** We thank the reviewer for pointing to us this relevant work on mini-batch versions of optimal transport (MBOT) and will add it to the camera ready version of the paper. MBOT is definitely a strong competitor to sliced versions of OT, but to our knowledge it has not been yet studied for measures living on manifolds, though we can expect some of its properties to be unaltered on this specific nature of data. We will also add [5] in which it is proposed to integrate over von Mises-Fisher distribution or mixture of von Mises-Fisher distributions instead of the uniform one.
> >
> >
> > [1] Jung, Sungkyu. "Geodesic projection of the von Mises–Fisher distribution for projection pursuit of directional data." Electronic Journal of Statistics 15.1 (2021): 984-1033.
> >
> > [2] Hauberg, Søren. "Directional statistics with the spherical normal distribution." 2018 21st International Conference on Information Fusion (FUSION). IEEE, 2018.
> >
> > [3] Galaz-Garcia, Fernando, et al. "Wrapped Distributions on homogeneous Riemannian manifolds." arXiv preprint arXiv:2204.09790 (2022).
> >
> > [4] De Cao, Nicola, and Wilker Aziz. "The power spherical distribution." arXiv preprint arXiv:2006.04437 (2020).
> >
> > [5] Nguyen, Khai, et al. "Improving relational regularized autoencoders with spherical sliced fused Gromov Wasserstein." arXiv preprint arXiv:2010.01787 (2020).

---

> > > ### Comment · Reviewer_yBhy · 2022-08-03
> > > **Response to authors**
> > >
> > > Thank you for your responses.
> > >
> > > From the result of the hyperspherical autoencoder, SSWAE seems to be only comparable to SWAE. Also, the FID scores are high compared to works in the literature. I guess that the CIFAR10 dataset does not have an underlining manifold which is a hypersphere. Could authors provide the reconstruction losses on the training set and the test set on reported datasets?
> > >
> > > Also, I wonder which application on real datasets that SSW can show the benefit of using the geodesic distance on the hypersphere.
> > >
> > > Minor: I am not sure if having the paper and the appendix in a single file violates the 9 pages limit of the rebuttal.
> > >
> > > I will ask if I have other questions.
> > >
> > > Best regards,

---

> > > > ### Author Response · Authors · 2022-08-04
> > > > **Answer to Reviewer yBhy**
> > > >
> > > > Thank you for your reply.
> > > >
> > > > For the results of FID, we believe that we cannot really compare with other works in the literature since we do not necessarily use the same architectures for the encoder and decoder. Here, we use the same architecture when we compare the methods and hence results are more comparable and it shows that SWAE and SSWAE give comparable results in term of FID.
> > > >
> > > > Note also that similar method which use sliced-Radon Sobolev instead of SWAE [1] report higher results than us in CIFAR10 (see Table 1, 120 for SWAE).
> > > >
> > > > We report here the reconstruction loss on training and validation set (averaged over 5 trainings):
> > > >
> > > > |Method\Dataset|MNIST|Fashion|CIFAR10|
> > > > |--------------|-----|-------|-------|
> > > > |SSWAE (Train) |4.8$\pm$ 0.04|6.67 $\pm$ 0.04|13.68 $\pm$ 0.11|
> > > > |SWAE (Train) | 4.96 $\pm$ 0.02 |6.96 $\pm$ 0.02|13.62 $\pm$ 0.04|
> > > > |SSWAE (Test) | 7.31 $\pm$ 0.04 | 8.01 $\pm$ 0.04 | 30.29 $\pm$ 0.15 |
> > > > |SWAE (Test) | 7.56 $\pm$ 0.07 | 8.25 $\pm$ 0.03 | 30.27 $\pm$ 0.09 |
> > > >
> > > > Real datasets on the sphere are often on $S^2$. We wanted to take advantage of the slicing process and of proposition 1. Hence, we focused on experiments in higher dimension  and with enforcing uniformity. We will look for applications on real datasets in future works.
> > > >
> > > > [1] Turinici, Gabriel. "Radon–Sobolev Variational Auto-Encoders." Neural Networks 141 (2021): 294-305.

---

### Official Review · Reviewer_JCLC · 2022-07-11

**Rating:** 5
**Confidence:** 4
**Soundness:** 3 good
**Presentation:** 3 good
**Contribution:** 3 good

**Summary:**

The paper proposed a new version of Sliced Wasserstein Distance on the sphere as a first step to deal with manifold data. Section 2 is to recall the definition of Wasserstein distance and Sliced Wasserstein distance. In Section 3, the authors introduced the spherical sliced Wasserstein distance, mainly based on the property that the sphere could be described by spherical coordinate system. They also defined the Radon transform for the SSWD and show some properties of Wasserstein distance on circle. The implementation is shown in Section 4. Section 5 is devoted for applications which include variational inference, generative modeling and density estimation.

**Questions:**

No

**Limitations:**

It is fine.

**Strengths And Weaknesses:**

Strengths: The idea of Spherical Sliced Wasserstein Distance is nice and interesting. The SSWD is well-defined and some of its properties are explored.

Weaknesses: The applications of SSWD shown in the paper are limited. In those experiments, the SSWD does not really outperform the SWD. The authors need to find applications with spherical data, where the advantages of SSWD could be shown clearly.

---

> ### Author Response · Authors · 2022-08-02
> **Response to Reviewer JCLC**
>
> We thank the reviewer for their comments. We address their concern below.
>
> **The applications of SSWD shown in the paper are limited.**
>
> While we agree that the results on the differents applications for SSW are only competitive with and not significantly outperforming SW, we would like to emphasize that SSW has many interests on its own, being a discrepancy that is intrinsic to the sphere, in the sense that we only use objects defined on the manifold. Hence, it is a first step toward defining geometric versions of SW on other manifolds which are not necessarily embedded in $\mathbb{R}^d$ and on which we could therefore not use SW.
>
> We also experimented on self-supervised learning on which we obtained fairly competitive results with state of the art contrastive methods. We will add in the revised version of the paper more comparisons between SW and SSW on this task. In particular, we observed that using SSW instead of SW in to enforce uniformity gives better performance with a better computational time using the closed-form provided in Proposition 1. We report on the following table the best performances obtained (by trying several regularization parameter $\lambda$ for both SW and SSW and denoting by $L$ then umber of projection).
>
>
> | Method | Encoder output | $S^2$ |
> |--------|----------------|-------|
> | _Wang et Isola._ [1] | 60.53 | 55.86 |
> | SW-SSL,  $\lambda = 1, L = 10$  | 62.65 | 57.77 |
> | SW-SSL,  $\lambda = 1, L = 3$   | 62.46 | 57.64  |
> | SSW-SSL, $\lambda = 20, L = 10$ | 64.89 | 58.91 |
> | SSW-SSL, $\lambda = 20, L = 3$  | 63.75 | 59.75 |
>
>
> We added these results in the revised version of the paper. And if accepted, we will move it to the main body of the paper.
>
> [1] Wang, Tongzhou, and Phillip Isola. "Understanding contrastive representation learning through alignment and uniformity on the hypersphere." International Conference on Machine Learning. PMLR (2020).

---

> > ### Comment · Reviewer_JCLC · 2022-08-09
> > **Response to the rebuttal**
> >
> > I would like to thank the authors for their replies. In summary, I like the theory, even it is not mathematically challenging to build up those theory. For the practical side, the theory needs good examples to demonstrate its advantages, which is not shown in the paper. Hence, I would like to keep my score unchanged.

---

### Official Review · Reviewer_ET9y · 2022-07-25

**Rating:** 4
**Confidence:** 2
**Soundness:** 3 good
**Presentation:** 2 fair
**Contribution:** 3 good

**Summary:**

The paper introduces a new formulation to compute the distance between probabilities distribution on the hyper-sphere, namely, the Spherical Sliced-Wasserstein (SSW). The proposed formulation involves a novel spherical Radon transform (to project to great circles) and the computation of the WD on the circle.
The method is validated on two tasks:  variational inference (matching a target distribution up to a scale) and autoencoders with a prior distribution on the spherical latent space.

**Questions:**

I would like the authors to address my previous two concerns:
- trying to add an intuitive layer to the mathematical formulation (possibly with supporting figures)
- expanding the analysis on the benefits of using a spherical distribution prior with autoencoder on datasets with different characteristics.

**Limitations:**

yes

**Strengths And Weaknesses:**

The paper is quite difficult to follow in the mathematical formulation without the proper background. If possible, it would be helpful to add some intuitive explanation or figure to better understand the main steps of the proposed formulation. Nevertheless, the paper tackles an interesting problem, and being able to compare density distributions on a hyperspherical domain could be useful for some machine learning tasks, as shown with the (toy) applications variational inference and autoencoders.

On this latter point, I found the experimental setup not properly explained. Most of the details are demanded of the supplementary material, which makes it difficult to understand what was done without jumping forth and back. It would also be nice to see more examples and analyses on the spherical latent space of the autoencoders (for instance, the different behaviour between datasets for which the data manifold is known to be spherical (e.g. rotations) and datasets the data manifold is presumed to be more complex or hierarchical).

---

> ### Author Response · Authors · 2022-08-02
> **Response to Reviewer ET9y**
>
> We thank the reviewer for his/her comments. Below, we address his/her questions and concerns.
>
> **Mathematical Background.** We agree that the mathematical background is quite demanding. To make the basic understanding of the method more intuitive, we propose to add in the next version of the paper a figure and/or a table on which we would compare the main ingredients of the classical SW distance and of SSW.
>
>
>
> | |Geodesics | Projection | Integration Set |
> |-|----------|------------|-----------------|
> |SW| Lines | $P^\theta(x) = \langle x, \theta\rangle$ | $S^{d-1}$ |
> | SSW | Great circles |$P^U(x)=\frac{U^Tx}{\|\|U^Tx\|\|_2}$ | $\mathbb{V}_{d,2}$ |
>
>
> **Expanding the analysis on the benefits of using a spherical distribution prior with autoencoders.**
>
> The benefit of using a spherical distribution prior with autoencoder was thoroughly studied in related works such as [1,2,3]. We agree that it would be interesting to study the latent space and the performance obtained on datasets with more complex structure, e.g. by using data which have a known spherical latent space (such as MNIST with rotated digits) . This is a nice idea that we will consider in further work.
>
> For existing datasets with complex structure (e.g. hierarchical),  it has been in part already done in several related works such as [4,5] which compare the performance obtained for different datasets and different latent spaces. Hence here, we focus on showing the ability of SSW to capture a nice latent space with a uniform prior for simple datasets such as MNIST and we expect to obtain similar behavior as in related work on datasets with more complex structure.
>
> [1] Davidson, Tim R., et al. "Hyperspherical variational auto-encoders." arXiv preprint arXiv:1804.00891 (2018).
>
> [2] Xu, Jiacheng, and Greg Durrett. "Spherical latent spaces for stable variational autoencoders." arXiv preprint arXiv:1808.10805 (2018).
>
> [3] Zhao, Deli, Jiapeng Zhu, and Bo Zhang. "Latent variables on spheres for autoencoders in high dimensions." arXiv preprint arXiv:1912.10233 (2019).
>
> [4] Grattarola, Daniele, Lorenzo Livi, and Cesare Alippi. "Adversarial autoencoders with constant-curvature latent manifolds." Applied Soft Computing 81 (2019): 105511.
>
> [5] Skopek, Ondrej, Octavian-Eugen Ganea, and Gary Bécigneul. "Mixed-curvature variational autoencoders." arXiv preprint arXiv:1911.08411 (2019).

---

> > ### Comment · Reviewer_ET9y · 2022-08-05
> > **Response to authors**
> >
> > For toy synthetic datasets that (partially) lie on spherical manifolds, you can look at the disentanglement literature (e.g. Locatello et al. http://proceedings.mlr.press/v119/locatello20a/locatello20a.pdf, or Fumero et al.  http://proceedings.mlr.press/v139/fumero21a/fumero21a.pdf), in particular, Shapes3D and Cars3D. Moreover, disentanglement, and in particular the later work of Fumero et al., could be an interesting practical application for the proposed latent space structure.

---

> > > ### Author Response · Authors · 2022-08-05
> > > **Response to Reviewer ET9y**
> > >
> > > We thank you for the references.
> > >
> > > We will look into these applications for further work.

---

### Author Response · Authors · 2022-08-02
**General Response**

We thank the reviewers for their positive and encouraging comments on our work. We added a revised version of the paper and we will sum up here the changes we have made.

The main issue raised by the reviewers is that experiments are not intensive enough. We recall here that our main objective is to propose **the first version of a sliced optimal transport divergence between measures supported on manifold**, in our case an hypersphere. Our contribution mostly lies in the construction of this divergence, and the associated properties, and less toward applications which would benefit from it. As such, our goal was to show that this divergence works in practice on selected applications that directly model data (or their representations) on hyperspheres.

 Nevertheless, we understand the need to show that our spherical sliced wasserstein (SSW) works better in practice than sliced wasserstein (SW) in the embedding space. We therefore conducted several complementary experiments: First, we added results on the Fashion MNIST dataset for the autoencoder experiment. Second, we completed some results that were available in the appendix of the document. In particular, we used SSW to prevent collapsing representations in self supervised learning, more specifically in the contrastive learning framework where the representations are projected on the sphere $S^2$.
The results obtained with SSL are competitive while slightly underperforming compared to other state of the art contrastive methods which enforce uniformity of the data (see Table 2) using an explicit interaction term between batch samples with a complexity of $O(dn^2)$. We added in the revision of the paper more comparisons using SW and SSW to enforce uniformity. In these experiments, not only does SW perform worse than SSW, but it also requires sampling and sorting a uniform distribution on the hypersphere to compute SW whereas the closed-form of SSW with the uniform distribution allows for a more efficient computation, **thanks to our new result in Proposition 1**.

| Method | Encoder output | $S^2$ |
|--------|----------------|-------|
| Supervised | 82.26 | 81.43 |
| SimCLR[1]     | **66.55** | 59.09 |
| _Wang and Isola._[2] | 60.53 | 55.86 |
| SW-SSL,  $\lambda = 1, L = 10$  | 62.65 | 57.77 |
| SW-SSL,  $\lambda = 1, L = 3$   | 62.46 | 57.64  |
| SSW-SSL, $\lambda = 20, L = 10$ | 64.89 | 58.91 |
| SSW-SSL, $\lambda = 20, L = 3$  | 63.75 | **59.75** |


If the paper is accepted, we will move some of these results in the main body of the paper as requested by one of the reviewers, as well as FID for the autoencoder experiment on CIFAR10 and CelebA.

We also would like the reviewers to take into account that one of the main interests of this work is to define a sliced-wasserstein based discrepancy which involves theoretically only objects which are well defined on manifolds, and we hope to pave the way towards defining such discrepancies on arbitrary manifolds, which are not necessarily embedded in Euclidean spaces, and on which the regular SW cannot be used.

[1] Chen, Ting, et al. "A simple framework for contrastive learning of visual representations." International conference on machine learning. PMLR (2020).

[2] Wang, Tongzhou, and Phillip Isola. "Understanding contrastive representation learning through alignment and uniformity on the hypersphere." International Conference on Machine Learning. PMLR (2020).

---

### Meta-Review · Area_Chair_Ce3A · 2022-08-26

**Recommendation:** Reject
**Confidence:** Certain

**Metareview:**

This paper has generated a long discussion and although it has strong theoretical merits, we all concord that the paper lacks of empirical motivations as well as a strong empirical evaluations with respect to distance distributions not exploiting manifold sructure and thosed define on a manifold.  Hence, we believe that at this point it would be preferable to have such empirical evidence (ideally with quantitative results on real-world problems) before accepting the paper.  Given that, we are sure that the paper will be much stronger and of broader interest to the ML community.

**Award:**

No

---

### Decision · Program_Chairs · 2022-09-14

Reject